# Cocrystals by Design: A Rational Coformer Selection Approach for Tackling the API Problems

**DOI:** 10.3390/pharmaceutics15041161

**Published:** 2023-04-06

**Authors:** Maan Singh, Harsh Barua, Vaskuri G. S. Sainaga Jyothi, Madhukiran R. Dhondale, Amritha G. Nambiar, Ashish K. Agrawal, Pradeep Kumar, Nalini R. Shastri, Dinesh Kumar

**Affiliations:** 1Pharmaceutical Solid State Research Laboratory, Department of Pharmaceutical Engineering and Technology, Indian Institute of Technology (Banaras Hindu University), Varanasi 221005, India; 2Solid State Pharmaceutical Cluster (SSPC), Science Foundation Ireland Research Centre for Pharmaceuticals, Bernal Institute, Department of Chemical Sciences, University of Limerick, V94T9PX Limerick, Ireland; 3Department of Pharmaceutics, National Institute of Pharmaceutical Education and Research, Hyderabad 500037, India; 4Wits Advanced Drug Delivery Platform Research Unit, Department of Pharmacy and Pharmacology, School of Therapeutic Sciences, Faculty of Health Sciences, University of the Witwatersrand, Johannesburg 2193, South Africa; 5Solid State Pharmaceutical Research, Hyderabad 500037, India

**Keywords:** coformer, cocrystal, solubility, stability, fumaric acid, oxalic acid, succinic acid, citric acid

## Abstract

Active pharmaceutical ingredients (API) with unfavorable physicochemical properties and stability present a significant challenge during their processing into final dosage forms. Cocrystallization of such APIs with suitable coformers is an efficient approach to mitigate the solubility and stability concerns. A considerable number of cocrystal-based products are currently being marketed and show an upward trend. However, to improve the API properties by cocrystallization, coformer selection plays a paramount role. Selection of suitable coformers not only improves the drug’s physicochemical properties but also improves the therapeutic effectiveness and reduces side effects. Numerous coformers have been used till date to prepare pharmaceutically acceptable cocrystals. The carboxylic acid-based coformers, such as fumaric acid, oxalic acid, succinic acid, and citric acid, are the most commonly used coformers in the currently marketed cocrystal-based products. Carboxylic acid-based coformers are capable of forming the hydrogen bond and contain smaller carbon chain with the APIs. This review summarizes the role of coformers in improving the physicochemical and pharmaceutical properties of APIs, and deeply explains the utility of afore-mentioned coformers in API cocrystal formation. The review concludes with a brief discussion on the patentability and regulatory issues related to pharmaceutical cocrystals.

## 1. Introduction

In pharmaceutical research and development portfolios, about 40% of commercialized APIs exhibit low water solubility. As stated by the biopharmaceutical classification system (BCS), drugs having low solubility and high permeability fall under class II [1]. One of the most critical challenges of BCS II drugs is to improve the solubility and dissolution rate [2]. In addition to poor aqueous solubility, most APIs exhibit undesirable physicochemical properties, flowability, compactability, etc., which hampers the solid dosage form development. The purity and performance of a drug product is severely impacted due to instabilities caused by polymorphic changes and degradation due to heat, light, and humidity during processing/storage [3,4]. Among the said problems, the poor aqueous solubility of active pharmaceutical ingredients (APIs) can be enhanced by micronization, amorphization, salt formation, cocrystallization, etc. [1]. Micronization, amorphization, and salt formation can improve the aqueous solubility of drugs, but the stability and processability are compromised in some cases [5,6]. The cocrystals approach has the potential to provide a safe way to improve solubility in addition to increasing/retaining the stability [7,8].

Cocrystals can be considered superior to amorphous API forms or solid dispersions because they possess the solubility advantages of high-energy solids and have a crystalline structure with good thermodynamic stability [9,10]. Cocrystals are defined by the European Medicines Agency (EMA) as “homogenous (single phase) crystalline structures made up of two or more components in a definite stoichiometric ratio where the arrangement in the crystal lattice is not based on ionic bonds (as with salts) and the components of a cocrystal may nevertheless be neutral as well as ionized” [11]. The United States Food and Drug Administration (USFDA) defines cocrystals as “Crystalline materials composed of two or more different molecules, typically API and cocrystal formers (coformers), in the same crystal lattice in a defined stoichiometric ratio.” Cocrystals are different from salts, polymorphs, solvates and hydrates [12]. The hydrogen-bonding interactions of API with the coformer alter its physicochemical properties and lead to enhanced pharmaceutical attributes [13]. Various methods [14] have been utilized till the present by researchers to prepare pharmaceutically acceptable cocrystals and have been illustrated in Figure 1.

Commercialized drug products provide some evidence of the efficacious application of cocrystallization in the pharmaceutical industry. Depakote^®^, Entresto^®^, Suglat^®^, Steglatro^®^, Lexapro^®^, ESIX-10^®^, Beta-chlor^®^, Cafcit^®^, Zafatek^®^, and Lamivudine/zidovudine Teva^®^, etc., are some commercially available pharmaceutical products that contain cocrystal-based APIs [15,16]. 

The coformer selection plays an important role in deciding the final cocrystal attributes. The coformers have the ability to modulate the API stability and solubility when prepared as a cocrystal by inducing changes in its crystal structure [1]. There have been a few studies that have reported deterioration of API properties after cocrystallization [3]. A variety of (GRAS) coformers generally regarded as safe are used to prepare pharmaceutically acceptable cocrystals [1,17,18,19]. The nature of the coformer used (acidic/basic/neutral) is known to influence the stability of the final cocrystal [1]. There have been a few instances wherein the cocrystallization technique was applied to improve the hygroscopic stability of moisture-sensitive drugs [20]. Other common instabilities such as hydrolysis, isomerization, photodegradation, etc., can also be effectively overcome by means of cocrystal preparation [21,22]. A large number of coformers with different functionalities have been used till present to prepare pharmaceutical cocrystals. The utility of chemicals as coformers depends upon the hydrogen-bonding ability of the molecules with the API. The good hydrogen-bonding strength and molecular geometry between the coformer and the API plays a vital role in the development of cocrystals [23]. According to the Etter rule, the hydrogen bond is formed if good hydrogen-bond donors and acceptors participate in the hydrogen bonding [24].

Currently available literature on coformer selection is predominantly focused on the mere cocrystal formation using different coformers. However, there is very little emphasis on selecting a coformer specifically to improve a particular aspect of an API [13,25,26,27]. In this review article, the authors discuss the coformer selection, their properties and impacts on enhancing a particular physicochemical property of an API. Authors have provided statistical analyses on most commonly used coformers. The properties, applications and recent reports on the usage of commonly used aliphatic carboxylic acid-based coformers, such as succinic acid, fumaric acid, oxalic acid, and citric acid, is discussed in detail. The marketed formulations based on these four coformers are discussed and their applicability in improving the API properties of all four coformers is compared. The patentability and regulatory factors governing the development of cocrystals is briefly discussed towards the end of the article.

## 2. Selection of Coformer

As mentioned earlier, coformers have a major role in the cocrystal development. The factors such as the type of functional group, pKa, their physical form, and their molecular size are to be considered during cocrystal formation using a particular coformer [28]. The coformer selection is primarily done by the experimental method and knowledge-based method. The experimental method is based on trial and error. Herein, an API is cocrystallized with empirically selected coformers and the formation of cocrystals is later confirmed by employing analytical techniques such as powder X-ray diffraction (PXRD), differential scanning calorimetry (DSC), etc. This method of cocrystal screening is thus very tedious and requires a huge amount of resources. Alternatively, various knowledge-based approaches can be put to use. Suitable coformers are being selected based on the hydrogen-bonding, pKa-based models, supramolecular synthon compatibility using the Cambridge Structure Database (CSD), lattice energy calculation, Hansen solubility parameter, thermal analysis, saturation temperature measurements, virtual cocrystal screening (using molecular electrostatic potential surfaces-MEPS), etc. [29,30]. The Hansen solubility parameter (HSP) compares the aqueous solubility of the coformer and the API, and the compounds with similar HSP have a higher probability of forming cocrystals [31]. The knowledge-based methods thus predict the formation of cocrystals even before experimentation, based on the structural features of API and coformer. Another method for selecting coformers is based on the supramolecular synthons. Supramolecular synthons are the structural units within the supermolecules that can be generated due to intermolecular interactions. Supramolecular synthons are of two types: supramolecular homosynthons with identical self-complementary functionalities and supramolecular heterosynthons with different but complementary functionalities. The heterosynthons are typically more durable. In general, amide homodimers and carboxylic acid heterosynthons are preferred [32,33]. Figure 2 shows some of the common supramolecular synthons occurring in the cocrystals.

The most preferred approaches for the selection of coformer and the generation of cocrystals are Cambridge Structural Database (CSD)-based screening, hydrogen-bond rules, and pKa rules, which are briefly discussed below.

### 2.1. Cambridge Structural Database (CSD)

It is possible to carry out supramolecular retrosynthetic analysis, which entails locating intermolecular units for the desired cocrystal structure, using the CSD [34]. The CSD contains crystallographic information regarding the hydrogen bonds formed between the drug and the coformer. Currently, the CSD repository contains over 1.2 million crystal structures [35,36]. Every entry in the CSD contains information on chemical structure and crystallographic data (such as space groups, lattice, symmetry, and crystal systems), crystal packing, molecular dimensions, molecular geometry, stereochemistry, structure representation, and conformational analysis [37]. Based on the understanding of geometries and preferred orientations of current intermolecular interactions, coformers can be chosen for cocrystallization with the APIs [34,38].

### 2.2. Hydrogen-Bond Rules

The hydrogen-bond rule is another approach for the selection of coformer. The hydrogen bond (X-H) is an attractive interaction between a hydrogen atom and an electronegative atom (X). Hydrogen bonding can occur within a molecule or between two different molecules [39]. The hydrogen bond rule provides valuable information about the favored hydrogen-bond selectivity, connectivity patterns, and stereo-electronic properties of hydrogen bonds for a specific functional group or combination of functional groups, in which hydrogen bonds are formed. There are generally three rules of hydrogen bond formation. The first rule was proposed by Donohue: all available acidic hydrogen in the molecular crystal structure of that compound will be used in hydrogen bonding. The second rule states that if hydrogen-bond donors are present, all good acceptors will be engaged in hydrogen bonding. According to the third rule, hydrogen bonds will form especially between the finest hydrogen-bond acceptor and the finest hydrogen-bond donor [40].

### 2.3. pKa Rule

The pKa difference between the acid-base pair can be estimated to predict whether the pair forms salt or cocrystal [41]. The acid dissociation constant (pKa) affects how well certain medications are absorbed orally. The BCS Class II drugs are further classified into Iia (acidic drugs), Iib (basic drugs), and Iic (neutral drugs) based on their pH-dependent solubility (and dissolution rate). Weakly acidic drugs with pKa ≤ 5 (flurbiprofen, ketoprofen, etc.) exhibit higher aqueous solubility at alkaline pH of the intestine and are classified as class Iia drugs. The weakly basic drugs with pKa ≥ 6 (carbamazepine, rifampicin, etc.), in contrast, exhibit higher aqueous solubility at acidic pH of the stomach and are classified as class Iib drugs. Drugs which do not exhibit pH-dependent solubility are classified into the class Iic category (neutral drugs such as danazol, fenofibrate, etc.) [18,42,43]. The formation of cocrystals and salt can be predicted by a study of the transfer of protons and is determined by the ∆pKa = [pKa (base) − pKa (acid)]. A pKa value difference of greater than 2 or 3 between the API and coformer symbolizes the transfer of proton between acid and base. The smaller pKa value difference (less than 0) indicates cocrystal formation whereas a large difference in the pKa values (≥ 2 or 3) indicates salt formation [13].

## 3. Coformer Impact on Pharmaceutical Attributes

Coformers, along with drugs, have an ability to alter the pharmaceutical attributes of a cocrystal system. Thus, when designing a cocrystal, it is crucial to consider the physicochemical properties of coformers as well. The applicability of coformers to improve the stability, mechanical properties, solubility, and permeability of APIs is discussed in the current section. Figure 3 and Figure 4 depict the different aspects of API and coformers that need to be considered during cocrystal preparation.

### 3.1. Role of Coformers in Solubility, Dissolution, and Bioavailability

Oral delivery of drugs is the most favored and patient-compliant method of drug administration in spite of other available routes such as parenteral, pulmonary, transdermal, etc. The oral route offers a painless method of drug administration with high acceptance that makes it the most convenient route. Most of the APIs can be formulated as an oral solid dosage at comparatively low cost, which in totality makes them an attractive avenue for patients and pharmaceutical companies alike. The cocrystal formation effectively improves the solubility of poorly water-soluble APIs, and the coformer selected has a major role to play in deciding the solubility of the cocrystal prepared. The physicochemical properties of coformers, such as solubility, ionization, etc., are to be considered while designing cocrystals of poorly water-soluble APIs. A few examples related to the mentioned coformer properties are discussed below.

Putra et al. [21] reported cocrystals of epalrestat with betaine in which they depicted a two-fold increase in solubility. The reason was attributed to the formation of a layered structure between drug and coformer, as depicted in Figure 5. As compared to epalrestat, betaine has higher water solubility with a higher tendency to go into solution, resulting in the formation of cocrystal with higher solubility. On contact with water, the rate of epalrestat dissolution is accelerated due to rapid dissolution of betaine, which was supported by the cocrystal having a 3.5-fold higher intrinsic dissolution rate compared to the parent drug (shown in Figure 6).

Selection of a coformer of intermediate solubility directly influences cocrystal solubility, as it will lead to a prolonged parachute effect. This hypothesis is effectively highlighted by furosemide and 2-picolinamide sesquihydrate cocrystal, as reported by Banik et al. [44]. Sustained super-saturation levels of dissolved drug were maintained over the period of 24 h. This was attributed to the fact that the coformer 2-picolinamide has intermediate solubility, which led to gradual leaching of the drug. Such a phenomenon was described by a new term, the ‘synthon-extended-spring-parachute effect’, and is depicted in Figure 7. Further, the cocrystal with higher polarity will have higher solubility and consequently higher dissolution compared to neutral ones. Surov et al. [45] reported non-ionic cocrystal of diclofenac with theophylline where only a 1.6-fold increase in solubility was observed, whereas Nugrahani et al. [46] reported zwitterionic cocrystal of the same drug with l-proline showing a 7.69-fold increase in solubility.

The solubility and dissolution improvement of API are correlated with coformer solubility. In some cases, this correlation is not followed, which could be due to the interplay of other factors. One of the most plausible reasons attributed to such behavior is the existence of a stronger interaction between drug and coformer, leading to the formation of differently behaving cocrystal. There have been numerous examples of such cases in the literature. One such case was reported by Aitipamula et al. [47], who found that although nicotinamide had higher solubility than theophylline, cocrystal of flufenamic acid with theophylline showed higher solubility and dissolution compared to one with nicotinamide due to the existence of a stronger intermolecular interaction between the drug and nicotinamide. The effect of cocrystal selection on solubility and dissolution rate is summarized in Table 1.

### 3.2. Role of Coformers in Improving the Mechanical Properties of Drug Molecule

The APIs must have optimal mechanical properties to ensure easy processing of unit operations like mixing, granulation, tableting, etc. The properties relating to plasticity, elastic recovery, tensile strength, presence of slip planes, attachment energy, etc., are significantly influenced by the crystal packing of molecules in API [3,57,58]. Introduction of a coformer in the API crystal structure can either enhance, deteriorate or retain the mechanical properties [3,4]. Therefore, the mechanical properties can be tuned by the proper selection of coformer. There are several reports where the incorporation of coformer leads to improved mechanical properties of APIs [59]. Sun et al. [59] improved the tabletability of caffeine by co-crystallizing with methyl gallate. The synthesis of cocrystal leads to improvements in the compaction properties due to the presence of slip planes in the structure of cocrystal. The planes are held by the weak van der Waals forces and can be easily overcome by applying pressure. The planes slide over each other, which improves the tabletability of the cocrystal form. Similarly, Karki et al. [58] and Ahmed et al. [60] improved the tabletability of paracetamol by preparing its cocrystals. Further, the bending flexibility of probenecid was transformed from a single-component system to the multiple-component system by choosing a coformer having symmetrical hydrogen bond donor/acceptor groups [61]. The bending flexibility of probenecid was also retained by introducing a molecular spacer coformer, as reported by Nath et al. [62]. This indicates that, based on design strategy, the mechanical properties of drug molecules can be altered. Other examples of coformers that influence the mechanical properties of drugs are provided in Table 2.

### 3.3. Role of Coformers in Stabilizing the Drug Molecule

The stability of a drug during processing and storage is the most important concern of the pharmaceutical industry. Any changes in the polymorphic form of an API leads to changes in its physicochemical properties and can deteriorate the quality and safety of the final product. APIs face physical stability issues, such as pseudopolymorphism, polymorphic transition, hygroscopicity, and chemical stability issues such as hydrolysis, photolysis, thermal degradation, chemical transformation, dimerization, etc. These instabilities are mainly related to the crystal structure of API molecules. Cocrystals are ideal to avoid these instabilities [70]. The depiction of utility of cocrystals in improving the stability of APIs is shown in Figure 8. A few research works aimed at improving the stability of API are reported below.

Andrew et al. [71] employed cocrystallization to inhibit the hydration of anhydrous caffeine. Cocrystals of caffeine were synthesized with acid coformers, such as oxalic acid, glutaric acid, maleic acid, and malonic acid. It was reported that the stability of the cocrystal towards hydration followed the order of strength of acid groups in coformers, where the oxalic acid cocrystal was stable while the glutaric acid, which has the weakest acid groups, showed the least stability. It was hypothesized that the driving force for stabilization of cocrystals was achieved by employing the hydrogen bond donor such as the carboxylic acid group for the basic imidazole nitrogen. This effectively prevents water incorporation into the lattice of a cocrystal. Similar studies were conducted with etoricoxib as well, where stability enhancement was seen due to the strong hydrogen bonding between the coformer and API [72]. Gao et al. [22] improved the chemical stability of adefovir dipivoxil by cocrystallizing with acidic and basic coformers, saccharin and nicotinamide, respectively. The results showed that the acidic coformer enhanced the stability while the basic coformer did not stabilize it to such an extent. The study authors hypothesized three reasons for this: (1) the acidic coformer provided a micro-acidic environment to the structure, inhibiting the hydrolysis, whereas the basic coformer enhanced the degradation; (2) the acidic coformer introduced into the crystal lattice inhibits the dimerization, which later may lead to degradation; (3) the strong hydrogen bond of an acidic coformer prevents the moisture attack on the functional groups of drugs which are prone to hydrolysis.

Instabilities due to the isomerization of API can also make them unstable. This mechanism of instability was controlled by a coformer in the cocrystal of epalrestat [21]. The reaction cavity in the cocrystal was decreased compared to the drug, leading to hindered molecular motion, and thus inhibited the isomerization. This leads to improved photostability of the cocrystal compared to the drug molecule. Similarly, stability improvement of various APIs was achieved by cocrystal formation with a coformer, as summarized in Table 3.

### 3.4. Role of Coformers in Enhancing the Permeability of Cocrystals

Oral administration of compounds is the preferred route for the administration of medicines owing to its well-established benefits. The majority of cocrystal synthesis is done with an objective to improve oral bioavailability of drugs posing problems of either low solubility or low permeability or both. Abundant literature is available to demonstrate solubility improvement by the cocrystallization technique. On the other hand, the problem of poor permeability still remains underexplored. Permeability depends on complex factors, such as transporters, gastric transit time, intestinal epithelial metabolism and many other in vivo factors. As summarized in Table 4, the available literature indicates most of the experiments for permeability evaluation have been conducted through Franz-diffusion cells employing an artificial membrane.

There are multiple reasons for permeability enhancement in cocrystals. An increase in the concentration gradient due to high supersaturation levels of drug in solution leads to the increased flux rate (rate of drug permeation) [54]. The heterosynthon interactions in cocrystals can also improve the permeability by increasing thermodynamic activity. This feature can be attributed to the reduced packing efficiency of heterosynthons compared to the homosynthons, and as result, the packing efficiency and density of crystals reduces. Another reason could be the use of relatively lipophilic coformers that enhance the partition coefficient of drugs after cocrystallization and improve the rate of permeation across the cell membrane [78]. In rare cases, the coformer used may inhibit various drug transporter proteins, such as P-glycoprotein (P-gp). P-gp are efflux proteins present in the apical region of the small intestine and involved in the out-transportation of absorbed drug molecules. Drugs such as caffeine, stearic acid, etc., can inhibit the P-gp transporters and increase the permeation of drugs [69,79].

Dai et al. [78] employed Franz-type diffusion cells to study the permeability of 5-fluorouracil, its cocrystals and corresponding physical mixtures through a silicon membrane. The studied cocrystals were 5-fluorouracil (5-FU) with 3-hydroxybenzoic acid (1), 4-aminobenzoic acid (2), and cinnamic acid (3). Cumulative amounts of cocrystal permeated per unit area (Q_n_) and steady penetration rate (J_s_) were higher than those of the drug (Figure 9A). 5-FU physical mixtures with 3-hydroxybenzoic acid (1PM) and cinnamic acid (3PM) showed similar Q_n_ while the physical mixture with 4-aminobenzoic acid (2PM) showed lesser cumulative diffusion compared to 5-FU (Figure 9B). The permeability of the 5-FU cocrystals was improved due to the replacement of the homosynthons formed between drug–drug by new drug–coformer heterosynthons. The weaker interaction between drug and coformer leads to rapid dissociation, which results in higher concentration of 5-FU cocrystals. Due to the higher concentration of 5-FU cocrystals, the flux rate increases, which leads to improved permeability.

Amaral et al. [79] reported cocrystal of dapsone with caffeine in which the permeability of drug, cocrystal and the corresponding physical mixture was evaluated in calu-3 human bronchial epithelial cells. The greater permeability in apical to basolateral direction was attributed to the complete dissociation of cocrystal and the absence of any crystal lattice interaction in the solution.

The discussed works do not consider factors such as transporter, gastric blood flow, metabolism inhibition, and beyond. Hence, there is a need to explore the role of coformer in improving permeability under the influence of the above-mentioned factors. In Table 4, instances of coformers enhancing the permeation of drugs are given. For better evaluation of permeability of cocrystals, there can be experimentation using the in vitro models that better correlate with an in vivo intestinal environment. In vitro methods consisting of artificial lipid membranes, such as parallel artificial membrane permeability assays (PAMPA) instead of cell-based such as Mardin–Darby canine kidney cells (MDCK), caco-2 cells, or tissue-based assays like intestinal membrane vesicles, would serve as better indicators of permeability improvement [80]. Figure 10 depicts the mechanism of permeability enhancement by cocrystals.

**Table 4 pharmaceutics-15-01161-t004:** Examples of studies reported on the effects of coformers on the permeability of drug molecules in cocrystals.

Drug	Coformer(s)	Model	Effect on Permeability	Mechanism	Reference
Dapsone (DAP)	Caffeine	Calu-3 human bronchial epithelial cells	Greater DAP permeability from apical to basolateral direction compared to DAP alone Reduction in efflux rate, i.e., basolateral to apical direction of DAP compared to pure DAP	Possible effect of membrane transporters by caffeine and its metabolite theophylline	[79]
Ethenzamide	2,4-dihydroxybenzoic acid	Diffusion apparatus with cellulose nitrate membrane	Improved permeability of cocrystals compared to pure ethenzamide	Higher solubility of cocrystal	[81]
Entacapone (ETP)	Acetamide Nicotinamide Isonicotinamide Pyrazinamide Isoniazid Theophylline	Diffusion apparatus with dialysis membrane-135	Cocrystals of ETP with theophylline and pyrazinamide exhibited higher diffusion rate compared to pure drug ETP-THP cocrystal hydrate had better diffusion rate compared to ETP-PYZ cocrystal	Higher solubility and higher permeability of the coformer (Theophylline BCS Class I and pyrazinamide BCS Class III)	[82]
1,2,4-Thiadiazole derivative	Vanillic acid	Franz-type diffusion apparatus with regenerated cellulose membrane	Increased flux with no change in apparent permeability coefficient values	Higher amount of dissolved drug leading to higher concentration gradient	[83]
Adefovir dipivoxil	Stearic acid	Caco-2 cells monolayers	Improved cell permeability	Stearic acid acts as P-gp inhibitor	[69]
Hydrochlorothiazide	Piperazine Picolinamide Tetramethyl pyrazine Isoniazid Malonamide	Franz diffusion apparatus with cellulose nitrate membrane	Higher cumulative amount permeated with high diffusion rate	Higher solubility leading to increased concentration gradient	[84]
			Lower cumulative amount permeated with marginal increase in initial flux		
			Marginal increase in diffusion rate with marginal increase in flux		
Furosemide	Anthranilamide 2,3,5,6-Tetramethylpyrazine Adenine Caffeine	Franz diffusion cell apparatus through a cellulose nitrate membrane	Higher cumulative amount permeated with high diffusion rate	Decrease in lattice energy with presence of weaker intermolecular interaction leading to increased solubility and ultimately higher concentration gradient	[44]
			Lower cumulative amount permeated with lower flux	High lattice energy contributing to poor solubility and lesser concentration in donor compartment	
5-Fluorouracil	4-Aminobenzoic acid 3-Hydroxybenzoic acid Cinnamic acid	Franz-type diffusion apparatus with a silicon membrane	Higher cumulative amount permeated with high diffusion rate	Presence of weaker intermolecular interaction between drug and coformer. Lipophilicity of coformer.	[78]

## 4. Coformers Reported in the Literature

The discussion in the earlier sections of this review explains the significance of coformer properties in deciding the final characteristics of a cocrystal. In this section, the authors present a list of coformers reported in the literature with different APIs in Table 5. The fully exhaustive list is provided in Appendix A. The following table can be used as a reference for further cocrystallization experiments.

## 5. Coformers Used in High Demand

In the last two decades, the demand for coformers has upscaled, as depicted in Figure 11. Urea, nicotinamide, benzoic acid, etc., are in high demand compared to other coformers. The utility of carboxylic acid-based coformers such as succinic acid, citric acid, fumaric acid, and oxalic acid has also increased marginally. The carboxylic acid-based coformers are discussed in detail in the upcoming section of the review article. 

## 6. Commercially Available Drug Products Based on Cocrystals

The ultimate aim of developing any technology/process is to make it reach the target population. Cocrystals play a vital role in improving the pharmaceutical properties of APIs. Commercialized cocrystal-based drug products are evidence of the efficacious application of cocrystallization in the pharmaceutical companies. Depakote^®^, Entresto^®^, Suglat^®^, Steglatro^®^, Lexapro^®^, ESIX-10^®^, Beta-chlor^®^, Cafcit^®^, Zafatek^®^, and Lamivudine/zidovudine Teva^®^, etc., are some commercially available pharmaceutical products that contain cocrystal-based APIs. The purpose of this section is to make the readers aware of the current scenario of the cocrystal technology. 

### 6.1. Depakote^®^

Depakote^®^ (other names Epilim, divalproex sodium, and Depakene) is used as an anti-epileptic agent that increases the level of gamma-aminobutyric acid. A depakote delayed-release tablet was approved by FDA in 1983, while ER was granted approval in 2002. Depakote contains valproic acid as an API and valproate sodium as a coformer. Valproic acid is in liquid form at room temperature, and sodium salt is highly hygroscopic. The cocrystal form of these two is less hygroscopic than the API itself [15,16,103,104,105,106].

### 6.2. Entresto^®^

Entresto^®^ contains sacubitril and valsartan as an API in the fixed-dose combination. The Entresto is used in the treatment of symptomatic heart failure and to reduce the risk of cardiovascular death. It is available in the form of a film-coated tablet containing sacubitril and valsartan: 24/26 mg and was approved by FDA in 2015. Entresto^®^ is a type of drug–drug cocrystal industrialized and marketed by Novartis, Basel, Switzerland. Valsartan is a neprilysin inhibitor and block angiotensin II receptor. Entresto^®^ is the best example of a drug–drug cocrystal for the improvement of the pharmacokinetics properties of API due to cocrystallization. Valsartan shows a bioavailability enhancement of 50% in Entresto^®^ compared to valsartan alone [15,16,107].

### 6.3. Suglat^®^

Suglat^®^ is effective against selective SGLT2 (Sodium-Glucose Co-Transporter 2) inhibitor used in the treatment of diabetes and is available in the form of a tablet. Astellas, Tokyo, Japan and Kotobuki Pharmaceutical Co., Ltd., Nishina, Shizuoka, Japan (“Kotobuki”) discovered Suglat through research collaboration. Suglat^®^ was launched by Astellus Pharma Inc. Tokyo, Japan and Kotobuki Pharmaceutical, Nishina, Shizuoka, Japan on 17 April 2014 in Tokyo. It is available in Suglat^®^ tablets 25 mg and 50 mg. It is a good example of a cocrystal-based product that contains ipragliflozin as an API and L-proline as a coformer. Ipragliflozin absorbs moisture and is converted to a hydrate form under storage conditions. The cocrystallization of ipragliflozin` with L-proline imparts stability against hydrate formation [15,16,105,108].

### 6.4. Steglatro^®^

Steglatro^®^ is indicated for treatment of insufficiently controlled type 2 diabetes mellitus in adults. It acts by inhibiting SGLT2. Steglatro^®^ contains ertugliflozin as an API and L-pyroglutamic acid as a coformer. The daily recommended starting dose is 5 mg. If more prominent glycemic control is required, the dose of ertugliflozin can be raised in individuals who tolerate 5 mg once daily to 15 mg once daily. It was approved by US FDA in 2017 and is marketed by Pfizer, New York, United States. Cocrystallization here serves the purpose of stability enhancement of the API, because ertugliflozin exists as an unstable amorphous material. In this, ertugliflozin and L-pyroglutamic acid are used in a 1:1 ratio to enhance the stability and physicochemical properties of ertugliflozin [15,105,109,110].

### 6.5. Lexapro^®^ & ESIX-10^®^

Lexapro^®^ contains escitalopram oxalate, which is a selective serotonin reuptake inhibitor (SSRI) used to manage and treat major depressive and generalized anxiety disorders. Escitalopram is a pure S-enantiomer of racemic citalopram, which is also an antidepressant medication. It was approved by US FDA in 2002 and is marketed by Allergan, Dublin, Ireland. A similar example of escitalopram oxalate cocrystals is ESIX-10. It is available in the market in tablet form (10 mg). It was approved in 2009 for the treatment of anxiety and depression [15,16,111,112].

### 6.6. Beta-Chlor^®^

The existence of chloral betaine as a cocrystal was discovered only recently, in 2016, though it was first chemically synthesized. Another example is chloral betaine (Beta-chlor^®^), which was afterward recognized as a cocrystal in 2016. Chloral hydrate was the first sedative that was chemically synthesized in 1832. This cocrystal was made up of betaine and chloral hydrate. The cocrystals improve the thermal stability of chloral betaine compared to the pure drug substance. The melting point of chloral hydrate is 60 °C, whereas the melting point of the prepared cocrystal of chloral betaine was reported to be 120 °C [15,105].

### 6.7. Cafcit^®^

Cafcit^®^ is another cocrystal that contains citrated caffeine or caffeine citrate. It is used to treat breathing problems in premature babies. Cafcit^®^ shows better dissolution behavior and exhibits lower hygroscopicity than caffeine. According to X-ray diffraction studies, the cocrystal is held together by O-H···N hydrogen bonds between citric acid’s carboxylic acid groups and caffeine’s imidazole moieties. The approval for the cocrystal was obtained by HIKMA pharmaceuticals in 1999 [16,105,113].

### 6.8. Zafatek^®^

Zafatek^®^ is a cocrystal-based tablet used as an anti-diabetes agent. It contains trelagliptin and succinic acid as API and coformer, respectively. Trelagliptin is an oral dipeptidyl peptidase IV inhibitor and was approved for use in Japan in March 2015. It is marketed by the Takeda pharmaceutical company, Tokyo, Japan [16,114].

### 6.9. Lamivudine/Zidovudine Teva^®^

Lamivudine-zidovudine cocrystal is the best example of a drug–drug cocrystal. It is indicated in antiretroviral combination therapy for the treatment of human immunodeficiency virus (HIV) infection. Lamivudine/zidovudine Teva is a generic product marketed by Teva pharmaceuticals. Zidovudine and lamivudine both have a number of hydrogen bond donor and acceptor groups. The cytosine fragment of lamivudine and the thymine fragment of zidovudine seem to be capable of forming synthons with substances that have complementary hydrogen-bonding groups [16,115,116,117].

### 6.10. Odomzo^®^

Odomzo^®^ contains sonidegib as an active ingredient for the treatment of basal cell carcinoma. Odomzo^®^ was approved by U.S. FDA in July 2015 and by EMA in August 2015. This is an example of cocrystals in which phosphoric acid is used as a coformer. The daily recommended dose is 200 mg of sonidegib, administered orally, and separated from the meal. Odomzo^®^ is available in capsule form and is manufactured by Sun Pharmaceutical Industries Ltd., Mumbai, India [118,119,120].

### 6.11. Mayzent^®^


Mayzent^®^ is used to treat multiple sclerosis. Mayzent^®^ contains siponimod as an API and fumaric acid as a coformer in a stoichiometric ratio of 2:1. Mayzent^®^ cocrystal is thermodynamically stable and is manufactured by Novartis, Basel, Switzerland. Mayzent^®^ was approved by U.S. FDA in 2019 and available in tablet form [121,122,123].

### 6.12. Seglentis^®^

Seglentis^®^ is a drug–drug cocrystal comprising celecoxib and tramadol. Seglentis^®^ was approved by U.S. FDA in 2021 and is manufactured by Kowa pharmaceuticals, Alabama, United States. It is used in the treatment of acute pain, and the daily recommended dose of Seglentis^®^ is 100 mg (56 mg celecoxib and 44 mg tramadol hydrochloride) [122,124].

### 6.13. Dimenhydrinate

Dimenhydrinate is the cocrystal of diphenhydramine (drug) and 8-chlorotheophylline (coformer). It was approved by U.S. FDA in 1982 and is manufactured by Watson Laboratories Inc, New Jersey, United States. Dimenhydrinate is available in tablet form (50 mg) for the treatment of motion sickness, including nausea and vomiting [17,125,126].

### 6.14. Ibrutinib

Ibrutinib, an anticancer medication used to treat chronic lymphocytic leukemia, was combined with fumaric acid to create a cocrystal that has better stability while exhibiting similar solubility to the original API. This cocrystal is still waiting for FDA clearance [15,127].

### 6.15. E-58425 (Clinical Trial Phase 3)

E-58425 is currently under phase 3 clinical trial for the treatment of severe acute post-operative pain. E-58425 is based on the cocrystallization technique to make the cocrystals of tramadol/celecoxib. This is an example of a drug–drug cocrystal in which two APIs are used in the formation of cocrystals. E-58425 exhibited superior analgesic activity compared to the tramadol and celecoxib combination. The synergistic effect of tramadol–celecoxib cocrystals was also reported by Manuel et al. in the treatment of severe acute post-operative pain. Sebastian et al. also reported that the co-crystal of tramadol–celecoxib (CTC) had a significant impact on efficacy in a phase 2 clinical trial [128,129,130,131,132].

### 6.16. TAK-020 (Clinical Trial Phase 1)

TAK-020 with Gentisic acid (coformer) is used in the treatment of rheumatoid arthritis. Currently TAK-020 is available in oral solution form. Takeda Pharmaceuticals, Tokyo, Japan is currently working on the TAK-020–gentisic acid cocrystals to form the tablet. TAK-020 is in phase 1 clinical trial. If Takeda Pharmaceuticals get positive outcomes from the clinical trials, it would be the first solid dosage form of the TAK-020. Kouya Kimoto also reported that cocrystals of TAK-020 with gentisic acid showed an enhanced dissolution rate [128,129,133,134]. All the above-mentioned cocrystal-based products are enlisted in Table 6.

## 7. Most Popular Coformers Utilized in Cocrystal-Based Marketed Formulations

On the basis of the available literature, aliphatic carboxylic acid-based coformers are the most commonly used in the marketed cocrystal preparations. These coformers exhibit favorable hydrogen-bonding interactions with the APIs, resulting in the formation of cocrystals. Though other coformers are reported to be used extensively in cocrystal research, the carboxylic acid-based coformers have surpassed all other coformers in terms of usage in marketed formulations. The reason for the usage of these coformers is not exactly known but they do possess a good number of hydrogen-bond donors and acceptors, which is an essential feature of cocrystal formation. Currently, the usage of coformers in marketed formulations is in the following order: fumaric acid, oxalic acid > succinic acid > citric acid. Additionally, a literature search was carried out in common search engines such as ScienceDirect, Web of Science and PubMed with the “name of coformer” followed by the word “cocrystals” as key words. The results are shown in Figure 12. From this search we can assess the current scenario of the scientific publications based on the mentioned coformers.

### 7.1. Fumaric Acid (FA)

The first instance of natural fumaric acid isolation was carried out from the plant fumaria officinalis. The other names of fumaric acid are trans*-*1,2-ethylenedicarboxylic acid or *(*E*)-*2-butenedioic acid; the term “fumarates” is also used synonymously. Chemically, fumaric acid can be synthesized from maleic anhydride. Fumaric acid is a colorless crystalline solid. The molecular formula of fumaric acid is C_4_H_4_O_4_. Fumaric acid is degraded by both aerobic and anaerobic microorganisms [136,137,138,139]. The chemical structure of fumaric acid is shown in Figure 13. The properties of fumaric acid along with other carboxylic acid-based coformers are tabulated in Table 7.

#### 7.1.1. Pharmaceutical Properties

Esters of fumaric acid such as mono and dimethyl fumarate have good pharmaceutical application in the treatment of multiple sclerosis and psoriasis. In 1994, DMF was initially made available on the market as Fumaderm^®^ [145].

#### 7.1.2. Fumaric Acid as a Coformer

Yang et al. [7] reported that fumaric acid can be a choice of coformer to form cocrystals. Dezhi et al. reported that fumaric acid has good water solubility compared to berberine chloride (BBC). BBC possess good pharmacological activities, but poor stability limits its applications. BBC–fumaric acid cocrystals improve the stability and dissolution rate compared to BBC alone. Similarly, cocrystals of promethazine hydrochloride with the fumaric acid as coformer in the ratio 2:1 possess good solubility and stability [146]. A few research works have demonstrated the use of fumaric acid as a coformer in improving the dissolution rate (approximately 6.1 × 10^−3^ mmol cm^−2^ min^−1^) of fluoxetine hydrochloride [147,148]. It is also reported that fumaric acid can effectively improve the therapeutic efficacy of the APIs by improving their physicochemical properties. Enoxacin is an anti-bacterial of the fluoroquinolone class having poor aqueous solubility. The cocrystal of enoxacin with fumaric acid enhanced the solubility and permeability of the drug and thereby improved its anti-bacterial activity as well [149,150,151]. The cocrystals of 6-nitroquinoline were grown using fumaric acid as a coformer in a 1:1 ratio by using the slow solvent evaporation method. The hydrogen bonds C–H···O and O–H···N in 6-nitroquinoline fumaric acid cocrystals stabilized the structure of the API [152]. Chaitanya et al. [153] reported that the use of fumaric acid as a coformer enhances the solubility, dissolution rate, and permeability of nicorandil. They also reported that nicorandil fumaric acid cocrystals have a good hardness property at lower compaction pressure. It is also reported that single crystals of L-histidinium can be synthesized by using fumaric acid as a coformer [154]. Similarly, another report shows that fumaric acid is capable of forming single crystals with L-phenylalanine. The obtained single cocrystal of L-phenylalanine has good thermal stability [155]. The hydrogen-bonding interaction of API with fumaric acid in cocrystals is depicted in Figure 14. 

According to the reported literature, fumaric acid used as a coformer in the formation of cocrystals plays a vital role. Fumaric acid showed the higher impact on the solubility and dissolution rate. Sildenafil–fumaric acid cocrystals obtained by the slow solvent evaporation method showed a great improvement in the solubility of nearly 5-fold compared to sildenafil alone. Not only did sildenafil cocrystals show improved solubility, but other cocrystals with fumaric acid also showed an increment in the solubility [156]. In the case of ketoconazole–fumaric acid cocrystals taken in the molar ratio 1:1, 1:2, and 1:3 made by using the slow solvent evaporation method, all three-molar ratio cocrystals showed improvement in the solubility, dissolution rate, and stability of the ketoconazole. The dissolution rate of ketoconazole–fumaric acid cocrystals was enhanced 1.65-fold compared to ketoconazole alone [157]. Cocrystals of promethazine hydrochloride with fumaric acid were prepared by mechanochemistry and slow solvent evaporation in the same molar ratio of 2:1. Both the promethazine hydrochloride–fumaric acid cocrystals have improved solubility and stability [146]. Below are the some reported cocrystals with fumaric acid as coformer showing improvement in solubility, dissolution rate, permeability, and stability. The utility of fumaric acid as coformer and its impact on the properties of prepared cocrystals is summarized in Table 8.

### 7.2. Oxalic Acid (OA)

Oxalic acid can be naturally obtained from bacteria, plants, fungi, and animals or can be chemically synthesized. Oxalic acid is odorless alpha, omega-dicarboxylic acid. The IUPAC name is ethanedioic acid and the formula is C_2_H_2_O_4_ [91,140,141]. The chemical structure of oxalic acid is shown in Figure 15. Refer to Table 7 for the other physical and chemical properties of the oxalic acid.

#### Oxalic Acid as a Coformer

From the earlier discussion, it is clear that a coformer plays a vital role in the formulation of cocrystal-based products. Oxalic acid is a coformer which has a good water solubility. Hrinova et al. [91] used oxalic acid as a coformer in the preparation of rivaroxaban cocrystals. Rivaroxaban belongs to BCS class II, having low solubility and high permeability. Oxalic acid is a highly water-soluble coformer that interacted with the rivaroxaban, forming the hydrogen bond. Rivaroxaban oxalic acid cocrystal not only increased the solubility but also showed significant improvement in the dissolution rate. Chen et al. [166] reported enhancement in solubility and bioavailability of apixaban cocrystal with oxalic acid as coformer. The prepared cocrystal performed better than the marketed product Eliquis^®^. Another report by Kusuma et al. [167] described improvement in stability of temozolomide which is an anti-cancer drug, marketed under the brand name Temodar^®^ or Temodal^®^. Temozolomide often changes in physical appearance from white to light tan/pink during storage. This discoloration is indicative of the degradation of temozolomide. Upon cocrystallization with oxalic acid, temozolomide showed better storage stability. Similarly, escitalopram oxalate–oxalic acid cocrystals are marketed under the trade name Lexapro^®^ in which oxalic acid is used as a coformer. The escitalopram oxalate has the stability issue, which is improved by oxalic acid after successful generation of escitalopram oxalate cocrystals [15,105]. Another report by Karki et al. [58] demonstrated that paracetamol (acetyl-para-aminophenol)–oxalic acid cocrystals are capable of improving the poor compressibility during tablet production. Chen et al. [168] reported that oxalic acid used as coformer in the preparation of xanthotoxin cocrystals enhanced its solubility, dissolution rate, and stability. The hydrogen-bonding interaction between the different APIs and oxalic acid is shown in Figure 16. 

Oxalic acid is also used as a coformer to enhance the solubility, dissolution rate, permeability, stability, and bioavailability of the drugs. According to some reported literature, there was no impact on the permeability of the drug, but there was a higher impact on the solubility on the drugs. The solubility of the drug increased up to 12-fold depending on the molar ratio, method of preparation, and other factors. Xanthotoxin–oxalic acid cocrystals showed a nearly 1.6-fold solubility improvement, while rebamipide–oxalic acid cocrystals showed a 7.29-fold solubility increment. Both cocrystals were made by liquid-assisted grinding, but there was a difference in the molar ratio of drug and oxalic acid. Rebamipide–oxalic acid cocrystals also showed an improvement in bioavailability [169]. It is also reported that apixaban–oxalic acid cocrystals taken in a molar ratio of 4:3 increased solubility approximately 2-fold and bioavailability 2.7-fold [166]. Table 9 summarizes the applications of oxalic acid as coformer in improving the physicochemical properties of APIs.

### 7.3. Succinic Acid (SA)

Georgius Agricola, a German chemist, discovered succinic acid (also known as butanedioic) with the molecular formula C_4_H_6_O_4_ [173]. SA is a dicarboxylic acid that exists as white, glittering crystals [174]. The chemical structure of succinic acid is shown in Figure 17. Other physical and chemical properties of succinic acid are given in Table 7.

The global demand for succinic acid is increasing and is approximately 30,000 tons per annum [175]. Market demand for SA is increasing tremendously, from USD 131.7 million in 2018 to USD 182.8 million in 2023 at a 6.8% CAGR (compound annual growth rate) [176].

Succinic acid has a good aqueous solubility that triggers the solubility and dissolution rate of APIs that belong to BCS class II. Succinic acid is commonly used as a coformer for the preparation of cocrystals [142]. Coformers are the backbone of cocrystal formation and of their solubility enhancement. Succinic acid is soluble in water (71 mg/mL) and is used as a coformer to improve the solubility of BCS class-II drugs. By introducing a more soluble coformer into the crystal lattice, which results in a lower solvation barrier, significant attempts have been undertaken in recent decades to increase the solubility, permeability, or bioavailability of poorly water-soluble medicines. The ultimate aim of using the coformer is to improve the solubility and dissolution rate of a poorly aqueous soluble drug by introducing the coformers into the BCS class-II drugs [18,92,142].

#### 7.3.1. Succinic Acid as a Coformer

Alhalaweh et al. [177] reported that urea-succinic acid cocrystals improve the solubility and thermodynamic stability of urea. In this work, urea as an API (e.g., to treat eczema & psoriasis) and succinic acid as a coformer was used to form the cocrystals. An acid–amide heterosynthon stabilized the 1:1 U-SA cocrystal, whereas amide–amide homosynthons and acid–amide heterosynthons stabilized the 2:1 cocrystals. The hydrogen bond interaction takes place between different APIs and succinic acid (coformer) to form the cocrystals, as shown in Figure 18.

In the case of itraconazole-succinic acid cocrystals, the dissolution rate and stability were increased compared to pure itraconazole. Two formulations were made by a liquid anti-solvent method (F1) and a gas anti-solvent method (F2). F1 achieved 50% drug release in 2 h while F2 achieved 92% drug release in 2 h. Carbamazepine–succinic acid cocrystals showed enhanced solubility, dissolution rate, stability and bioavailability compared to pure carbamazepine. Furthermore, fluoxetine–succinic acid cocrystals were made by using the slow solvent evaporation method in which fluoxetine–succinic acid was taken in the molar ratio 2:1. In this case, only the solubility of fluoxetine increased, approximately 2-fold. Table 10 below summarizes the role of succinic acid in enhancing the solubility, dissolution rate, bioavailability and stability of drugs.

#### 7.3.2. Other Applications

Succinates (most commonly calcium succinate, potassium succinate, and sodium succinate) are extremely helpful in the treatment of long-term illnesses and injuries. These are typically employed medically as sedatives, antispasmodics, antirheumatics, and contraceptives. Succinic acid is also employed as an antioxidant and a potassium ion inhibitor. Succinic acid is also a useful product for athletes. As a result, the dicarboxylate could be considered as an “elixir of youth” [96].

### 7.4. Citric Acid

Citric acid is a colorless, odorless white crystalline powder; its chemical name is 2-hydroxypropane-1,2,3-tricarboxylic acid. The molecular formula of citric acid anhydrate is C_6_H_8_O_7_. The other physical and chemical properties of the citric acid are mentioned in the Table 7. Lemon, orange, pineapple, strawberry, red currant, cranberry, and other fruits mostly contain citric acid. In 2021, the volume of the global citric acid market was 2.7 million tons. By 2027, the market is anticipated to grow to 3.2 million tons [187,188]. The chemical structure of citric acid is shown in Figure 19.

#### Citric Acid as a Coformer

There are various studies showing the successful applicability of citric acid as a coformer in increasing the solubility of poorly aqueous soluble drugs by many times. Yan et al. [143] reported that the solubility of metformin HCl was increased by using citric acid as a coformer. A similar example of berberine chloride was studied by Lu et al. [6] who reported that berberine chloride shows stability issues during wet granulation for tablet production. Cocrystals of berberine chloride with citric acid as a coformer were more stable compared to berberine chloride alone. A study by Hsu et al. [189] reported that the stability of theophylline improved after preparation of its cocrystal with citric acid. Deng et al. [190] reported that dapagliflozin possess the stability problem at high temperature and also has hygroscopicity issues. Cocrystals of dapagliflozin made by the use of citric acid (coformer) improved the stability of dapagliflozin. Furthermore, Wang et al. [56] described in their research that pyrazinamide (an anti-tuberculosis drug) belongs to BCS class II and has a solubility problem. Cocrystals of pyrazinamide with citric acid enhanced both the solubility and dissolution rate compared to pyrazinamide alone. Additionally, norfloxacin-citric acid cocrystals showed improved solubility compared to norfloxacin alone [191]. Another report by Revika et al. [192] described that ethyl p-methoxycinnamate used as anti-inflammatory agent showed a 44.19% increase in solubility in its cocrystal form compared to ethyl p-methoxycinnamate alone. Fahad et al. [193] reported that cocrystals of simvastatin in which citric acid was used as a coformer showed greater solubility, dissolution rate, and bioavailability. The hydrogen-bonding interaction between the different APIs and citric acid is shown in Figure 20.

Metformin hydrochloride–citric acid cocrystals obtained by solution crystallization, neat grinding, and liquid-assisted grinding in the ratio of 1:1 increased its solubility by 1–4-fold. The bioavailability enhancement of metformin cocrystals was seen as well. Furthermore, rebamipide–citric acid cocrystals exhibited 12.58-fold improvement in solubility compared to rebamipide alone. The intrinsic dissolution rate of the cocrystal was ~13.2 times higher than the API alone [169]. It is essential to consider the molar ratio of API:coformer taken during the cocrystallization experiments [194]. For example, simvastatin–citric acid required a molar ratio of 1:1 to form cocrystals having improved the solubility profile. Conversely, nefiracetam–citric acid cocrystals required an API:coformer ratio of 2:1 to show improvement in the solubility. Various other examples have been summarized in Table 11.

## 8. Comparison of Coformers

On the basis of scientific papers and the literature trend, four coformers are selected: fumaric acid, oxalic acid, succinic acid, and citric acid. All the coformers used in the cocrystals showed the improvement in the solubility, dissolution rate, bioavailability, and stability, while only fumaric acid also showed the permeability enhancement. Table 12 summarizes the role of fumaric acid, oxalic acid, succinic acid, and citric acid in improving the physicochemical properties of drug.

The number of commercialized products comprised of fumaric acid or oxalic acid as a coformer are higher compared to the other two coformers. The scientific literature and approved commercialized products indicate the successful application of fumaric and oxalic acid as coformers in generating cocrystals of problematic APIs. However, there is a need for studies exploring the applicability of any particular coformer in improving the physicochemical properties of a particular class of APIs. The cocrystal formation is principally governed by the intermolecular hydrogen-bonding interactions and the changes these interactions bring about in the crystal structure. Hence, further research is required to support the claim of fumaric and oxalic acid as the best-suited coformer for API cocrystallization. Based on currently available marketed cocrystal-based formulations, it can be said that the carboxylic acid-based compounds (fumaric acid and oxalic acid in particular) can be best suited for cocrystallization to improve pharmaceutical properties. However, the authors feel that industrial input is essential to prove the potential of any coformer.

## 9. Patentability Issue Criteria and Regulatory Guidelines of Pharmaceutical Cocrystals

Figure 21 shows the major steps involved in the development of cocrystal-based formulations. After the successful development of a cocrystal-based product, the next stage is to obtain the necessary approvals from the concerned regulatory bodies for commercialization of the developed product. Hence, this section briefly explains the current regulatory scenario with respect to product patenting and filing. 

It is necessary to improve the regulatory procedure for the filing and granting of patent as well as for the regulatory approval to commercialize the product. There are generally three conditions for the granting of patent, such as novelty, non-obviousness and utility. There are two ways to file the patent application in patent office: either by national phase application or an international phase application (PCT route). The guidelines for the pharmaceutical cocrystals were first published by the USFDA in 2013. As per the guideline, pharmaceutical cocrystals are considered to be a drug product intermediate that requires additional regulation. In the latest guidelines from USFDA in 2018, cocrystals were included as a drug substance and defined as “crystalline materials composed of two or more different molecules, one of which is the API, in a defined stoichiometric ratio within the same crystal lattice that are associated by nonionic and noncovalent bonds”. The USFDA also stated that a coformer is the component that interacts non-ionically with the API in the crystal lattice, that is not a solvent (including water), and is typically nonvolatile. For the regulatory approval of pharmaceutical cocrystals, there are two possibilities: the new drug application (NDA) pathway (505(b)(2)), and the abbreviated new drug application (ANDA) pathway (505(j)). The condition for an NDA application is that the cocrystals not have an active pharmaceutical ingredient that is already a reference listed drug (RLD) [122]. On the other hand, the applicant can file an ANDA application for the cocrystals which contains the previously approved drug (RLD). Mayzent is the best example for the newly approved cocrystals through the NDA route because its active pharmaceutical ingredient was not mentioned in the RLD [12,122,129,199].

## 10. Conclusions

In the current scenario, poor solubility, poor dissolution rate, poor permeability, low bioavailability, and instability are the major reasons for the failure of an active pharmaceutical ingredient (API). The researchers are focusing on mitigating these APIs issues. Cocrystallization is a proven approach to enhance the physicochemical properties of APIs and thus overcome the problems associated with APIs. Coformers play a paramount role in cocrystallization, as the final properties of cocrystal are dependent on the coformer characteristics and its interaction with the API. Research works have been presented wherein the cocrystallization experiments have resulted in improvement of the earlier-mentioned API properties. However, there is not enough evidence to prove the usage of any particular coformer to solve all API issues. A coformer must be selected based on the nature of the interaction with the API and the positive changes it brings about in the crystal structure after association. On the basis of commercialization potential, the aliphatic carboxylic acid-based coformers have gained prominence. Almost six commercialized cocrystal-based products are based on fumaric acid (two products), oxalic acid (two products), succinic acid (one product), and citric acid (one product). A few research works have been presented in this article, which shows the nature of the association of these coformers with the API and their utility in cocrystallization experiments. The ongoing research on these coformers indicates that their utility is still under exploration, which is a positive indication for cocrystallization-based research. Moreover, a small but important section related to the patentability and regulations concerning cocrystals is presented. Herein, the possible routes for filing a patent or product are discussed. Finally, the authors urge the readers/researchers to consider the carboxylic acid-based coformers for cocrystallization experiments so as to obtain concrete information on their utility as ultimate coformers. 

## Figures and Tables

**Figure 1 pharmaceutics-15-01161-f001:**
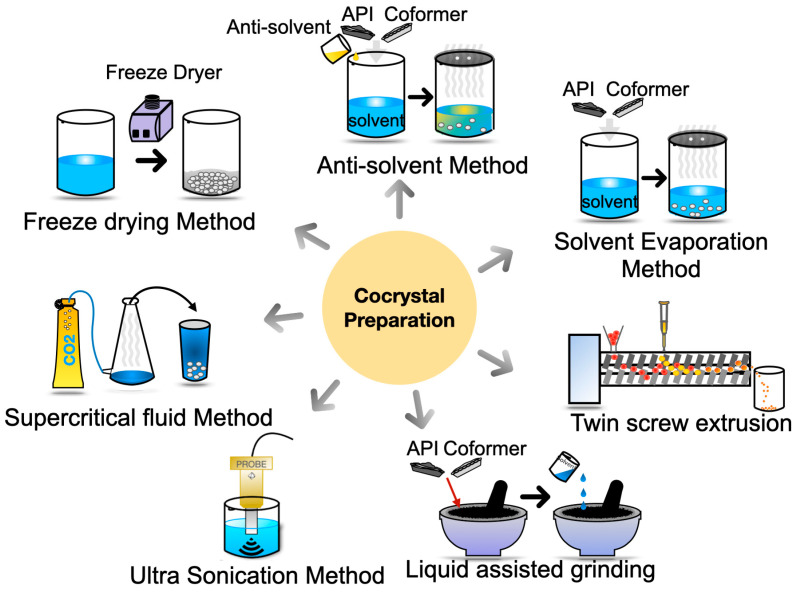
Techniques utilized for cocrystal preparation.

**Figure 2 pharmaceutics-15-01161-f002:**
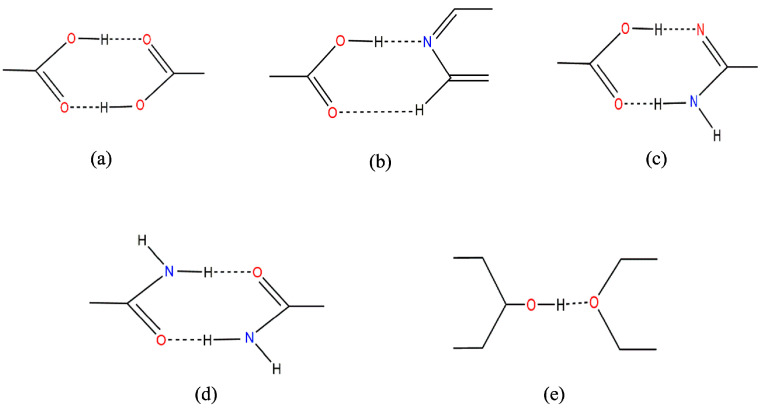
The most common supramolecular synthons in crystal engineering are (**a**) carboxylic acid dimer homosynthon, (**b**) carboxylic acid and pyridine group heterosynthon, (**c**) amide dimer heterosynthon, (**d**) amide group heterosynthons, and (**e**) alcohol and ether group heterosynthons.

**Figure 3 pharmaceutics-15-01161-f003:**
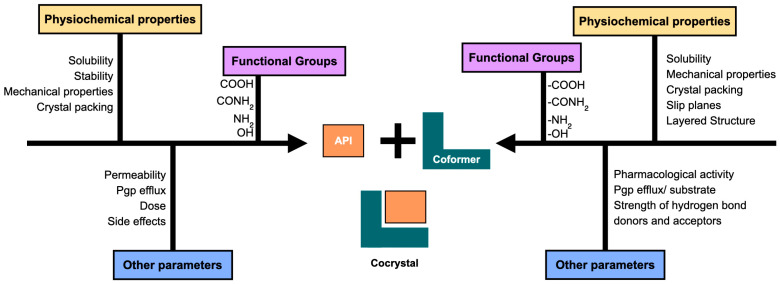
Properties of coformer and drug to be considered while designing the cocrystal.

**Figure 4 pharmaceutics-15-01161-f004:**
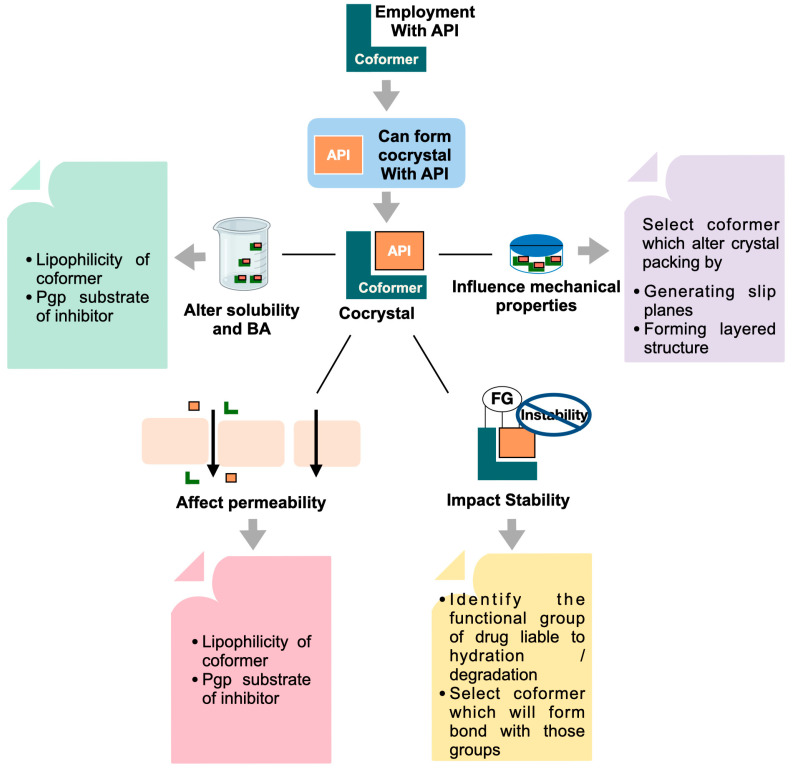
Pharmaceutical attributes affected by cocrystal formation and its relation to coformer properties.

**Figure 5 pharmaceutics-15-01161-f005:**
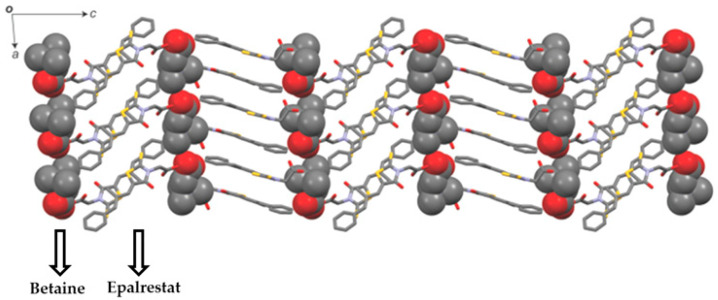
Layered cocrystal structure of epalrestat and betaine. Reprinted (adapted) with permission from [21]. Copyright (2018) American Chemical Society.

**Figure 6 pharmaceutics-15-01161-f006:**
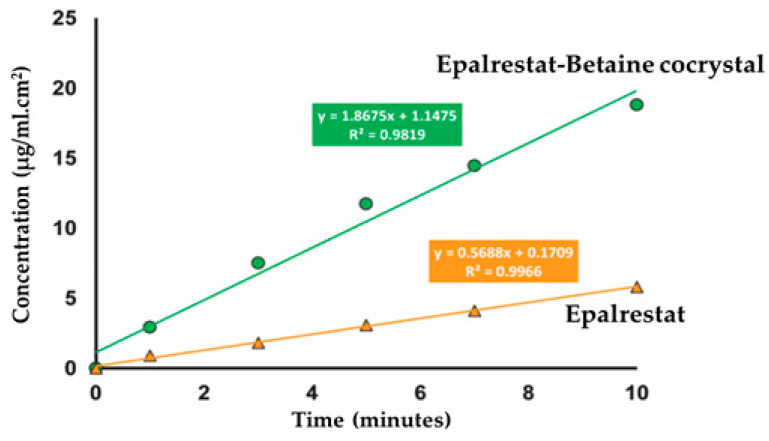
Intrinsic dissolution rate experiment results of epalrestat and its cocrystal. Reprinted (adapted) with permission from [21]. Copyright (2018) American Chemical Society.

**Figure 7 pharmaceutics-15-01161-f007:**
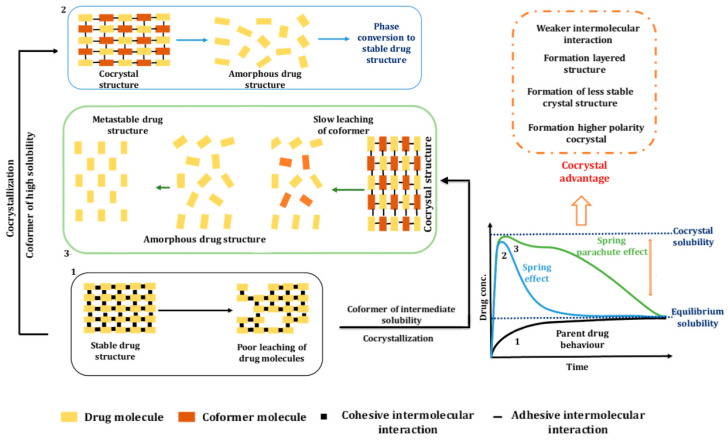
Illustration of the influence of coformer solubility on the spring and parachute effect of cocrystals.

**Figure 8 pharmaceutics-15-01161-f008:**
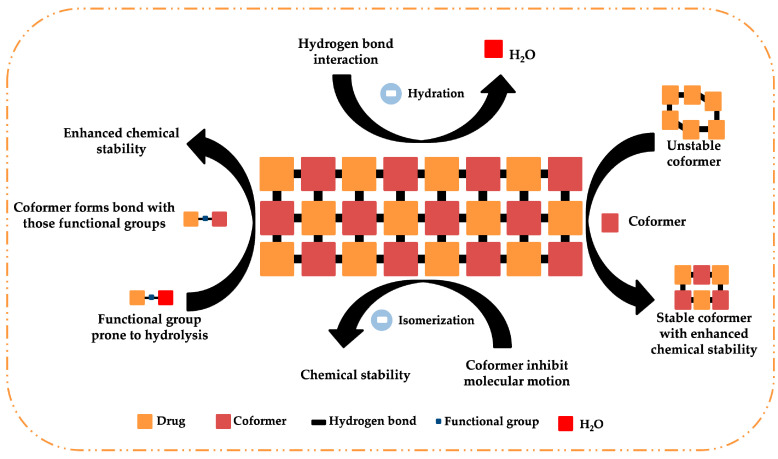
Role of coformers in stabilizing the drug molecule.

**Figure 9 pharmaceutics-15-01161-f009:**
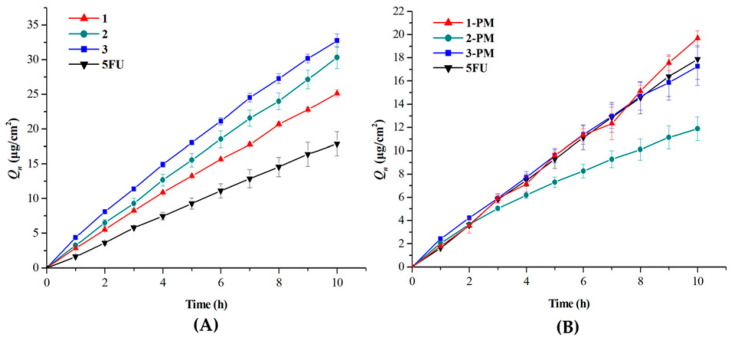
Cumulative amount permeated for (**A**) Cocrystal, (**B**) Physical mixtures of 5-fluorouracil. Reprinted (adapted) with permission from [78]. Copyright (2016) American Chemical Society.

**Figure 10 pharmaceutics-15-01161-f010:**
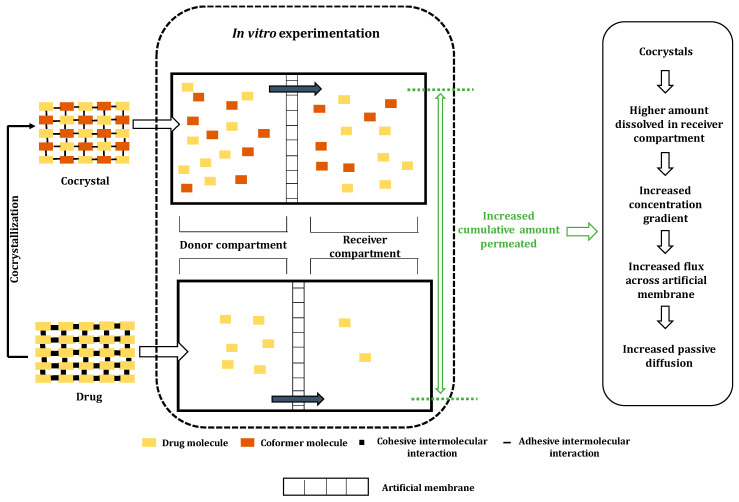
Illustration of enhanced permeability of cocrystals in comparison to pure drug.

**Figure 11 pharmaceutics-15-01161-f011:**
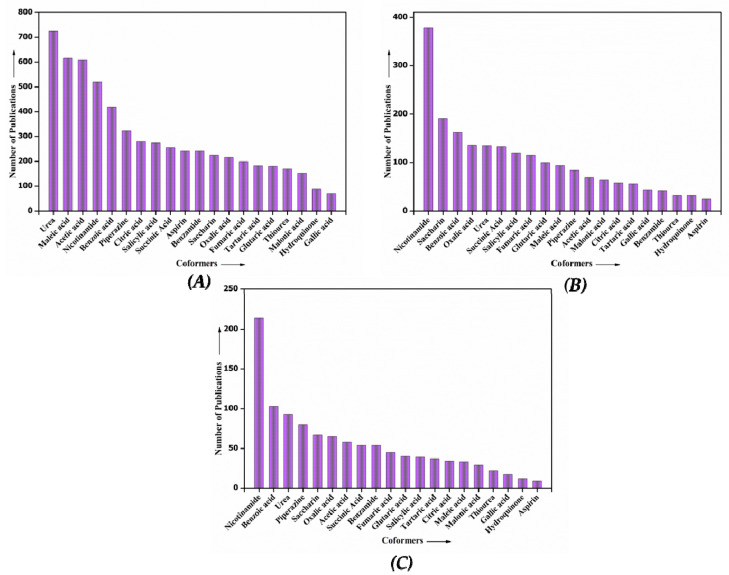
Graphical representation of coformers in high demand in the last 20 years by (**A**) ScienceDirect (**B**) Web of Science, and (**C**) PubMed.

**Figure 12 pharmaceutics-15-01161-f012:**
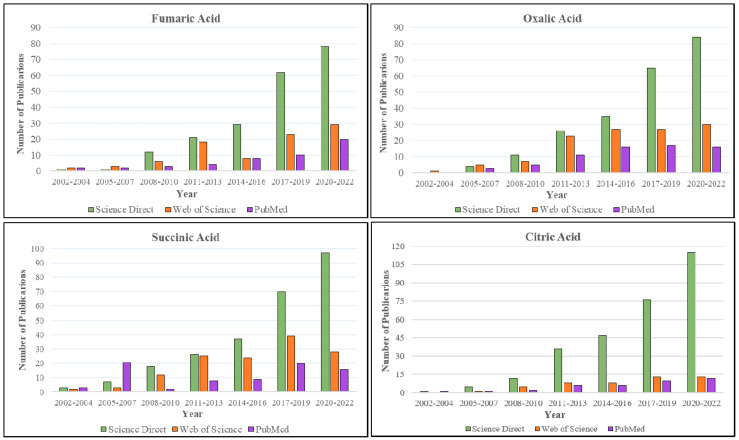
Literature trend of carboxylic acid-based coformers usage in preparation of cocrystals.

**Figure 13 pharmaceutics-15-01161-f013:**
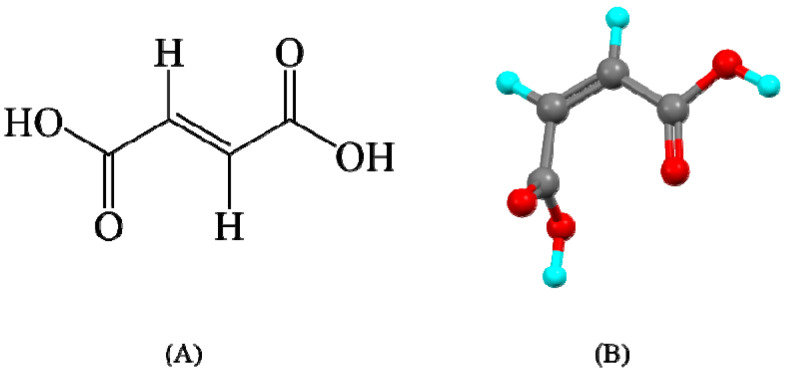
Chemical structure of fumaric acid (**A**) 2d and (**B**) 3d.

**Figure 14 pharmaceutics-15-01161-f014:**
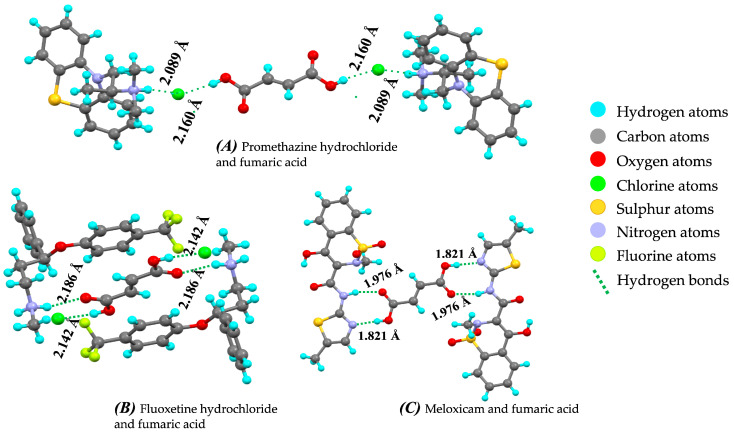
Hydrogen bond interaction and bond length between drug and coformer represented as (**A**) Promethazine hydrochloride and fumaric acid (#1853757), (**B**) Fluoxetine hydrochloride and fumaric acid (#254849), and (**C**) Meloxicam and fumaric acid (#819114) (from Mercury 2022.2.0 (Build 353591)). #—CCDC Identifier number.

**Figure 15 pharmaceutics-15-01161-f015:**
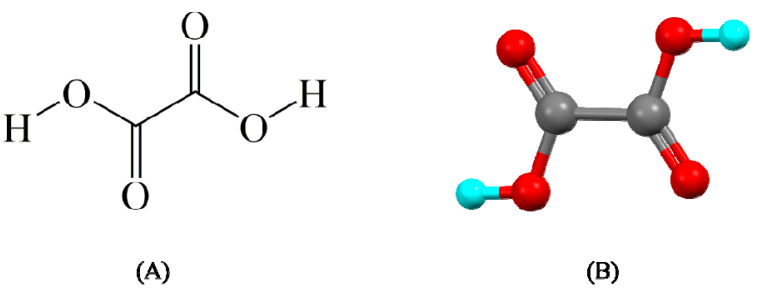
Chemical structure of oxalic acid (**A**) 2d and (**B**) 3d.

**Figure 16 pharmaceutics-15-01161-f016:**
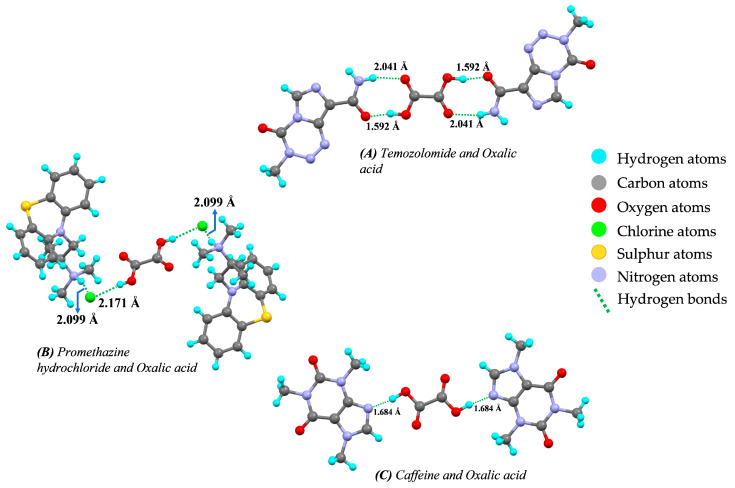
Hydrogen-bond interaction and bond length between drug and coformer represented as (**A**) Temozolomide and oxalic acid (#881197), (**B**) Promethazine hydrochloride and oxalic acid (#1853755), and (**C**) Caffeine and oxalic acid (#272620) (from Mercury 2022.2.0 (Build 353591)). #—CCDC Identifier number.

**Figure 17 pharmaceutics-15-01161-f017:**
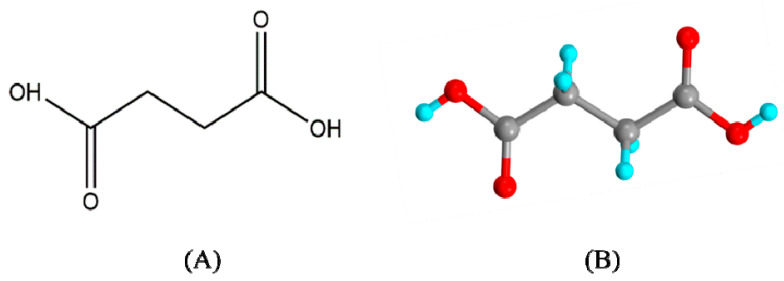
(**A**) Chemical structure of succinic acid (**A**) 2d and (**B**) 3d [175].

**Figure 18 pharmaceutics-15-01161-f018:**
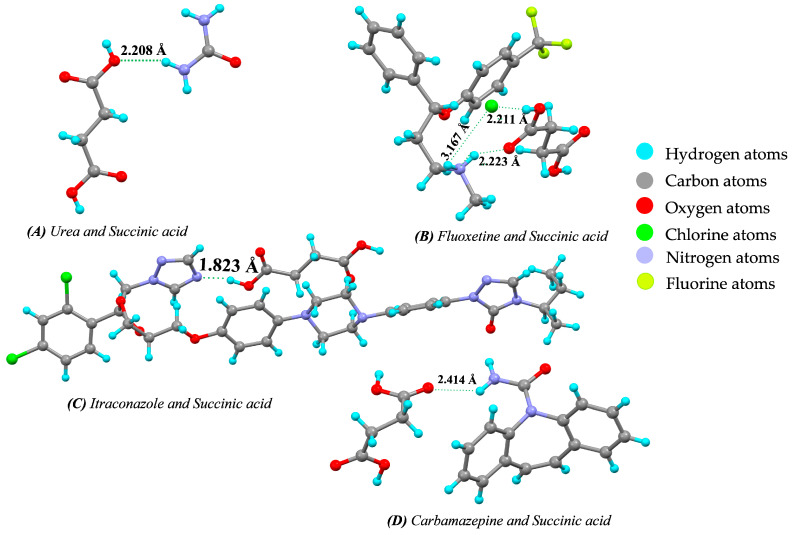
Hydrogen bond interaction and bond length between drug and coformer, represented as (**A**) Urea and Succinic acid (#795527), (**B**) Fluoxetine and Succinic acid (#254848), (**C**) Itraconazole and Succinic acid (#218567), and (**D**) Carbamazepine and Succinic acid (#671498) (from Mercury 2022.2.0 (Build 353591)). #—CCDC Identifier number.

**Figure 19 pharmaceutics-15-01161-f019:**
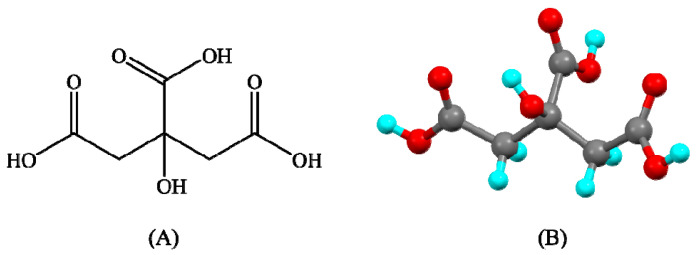
Chemical structure of oxalic acid (**A**) 2d and (**B**) 3d.

**Figure 20 pharmaceutics-15-01161-f020:**
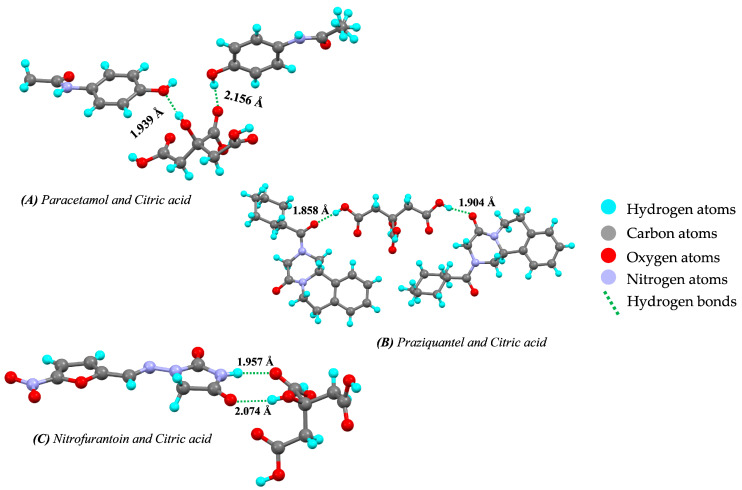
Hydrogen bond interaction and bond length between drug and coformer represented as (**A**) Paracetamol and citric acid (#803736), (**B**) Praziquantel and citric acid (#2094551), and (**C**) Nitrofurantoin and citric acid (#835423) (from Mercury 2022.2.0 (Build 353591)). #—CCDC Identifier number.

**Figure 21 pharmaceutics-15-01161-f021:**
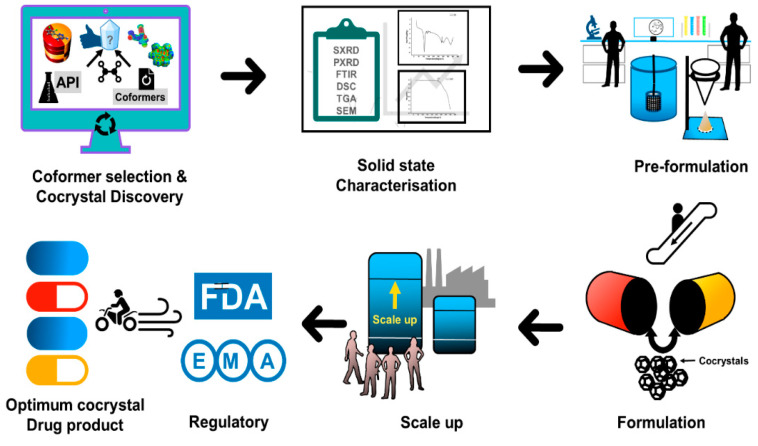
An illustration of steps involved in cocrystal-based product development.

**Table 1 pharmaceutics-15-01161-t001:** Examples of studies reported on the effects of coformers on the solubility, dissolution rate, and bioavailability of drug molecules in cocrystals.

Drug	Coformer	Effect on Solubility (Comparison with Drug)	Effect on Dissolution (Comparison with Drug)	Mechanism	Reference
Epalrestat	Betaine	2-fold increase	3.5-fold increase	Presence of layered structure with alternate drug and coformer	[21]
Glibenclamide	Hippuric acid	2.2-fold increase	2.2-fold increase	Hydrogen-bonding interactions between drug and coformers Melting point of cocrystal Solubility of coformer	[48]
	Nicotinic acid	3-fold increase	3-fold increase		
	Theophylline	1.5-fold increase	1.6-fold increase		
	Succinic acid	3.5-fold increase	3.7-fold increase		
Diclofenac	l-proline	7.69-fold increase	-	Formation of layered structure	[47]
Resveratrol	Piperazine	2.48-fold increase	-	Hydrogen bond network	[49]
	Methenamine	1.46-fold increase			
Felodipine	Glutaric acid	-	2.41-fold increase at pH 1.2	Lowering of melting point	[50]
Furosemide	Isonicotinamide	5.6-fold increase (Apparent Solubility-Phase transformation)	No change	Coformer solubility	[51]
Eprosartan mesylate	Salicylic acid	6-fold increase	Increased IDR	Coformer solubility Hydrogen-bonding interaction	[52]
	p-aminobenzoic acid	32-fold increase			
	Succinic acid	61-fold increase			
Acyclovir	Fumaric acid	5.5-fold increase	Increased dissolution rate	Coformer solubility Melting points of cocrystal	[53]
	Malonic acid	5.7-fold increase	Increased dissolution rate		
	DL-tartaric acid	5.3-fold increase	Increased dissolution rate		
Hydrochlorothiazide	Nicotinic acid	0.72 decrease	-	Formation of higher polarity cocrystals	[54]
	Nicotinamide	1.27 increase			
	4-aminobenzoic acid	0.21 decrease			
	Succinamide	2.4-fold increase			
	Resorcinol	1.3-fold increase			
Itraconazole	Aspartic acid	3.2-fold increase	Improved dissolution rate	Hydrogen-bonding interaction	[55]
	Proline	2.2-fold increase			
	Serine	2.5-fold increase			
	Glycine	2.3-fold increase			
	Succinic acid	1.6-fold increase			
Pyrazinamide	Adipic acid	0.7-fold decrease	0.6-fold decrease	Hydrogen-bonding interaction	[56]
	Sebacic acid	0.7-fold decrease	0.4-fold decrease		
	trans-Aconitic acid	1.6-fold increase	1.8-fold increase		
	Citric acid	1.2-fold increase	1.4-fold increase		

- Not mentioned.

**Table 2 pharmaceutics-15-01161-t002:** Examples of studies reported on mechanical properties of drug molecules in cocrystals.

Drug	Coformer	Issue of Drug/Objective of the Study	Parameter Improved	Inference	Reference
Caffeine	Methyl gallate	Severe lamination and over compaction	Powder compaction and tensile strength	Presence of slip planes resulted in improved compaction properties	[59]
Paracetamol form II	Oxalic acid Theophylline Naphthalene Phenazine	Poor tablet-forming ability	Tensile strength, elastic constant and lattice energies, elastic compliance tensor	Layered structure of cocrystals leads to superior tablet formation ability	[58]
Theophylline	Methyl gallate	To examine the effects of cocrystallization on crystal mechanical properties	Elastic modulus, indentation values, crystal slip planes and Burger’s vector	Tableting performance theophylline > co-crystal > methyl gallate	[57]
Ibuprofen and Flurbiprofen	Nicotinamide	To demonstrate improvement of pharmaceutical properties over pure drug crystal	Powder compaction analysis (tensile strength)	Tabletability of cocrystal is apparently higher due to its higher bonding strength.	[63]
Paracetamol	Trimethyl glycine	Poor tablet-forming ability	Hardness and particle-size distribution	Improved compression properties due to structural stability and changed crystal face	[64]
Voriconazole	Fumaric acid 4-hydroxybenzoic acid 4-aminobenzoic acid Oxalate salt	Too soft for tableting and compacting	Nanoindentation, elastic modulus and hardness	Hardness improved in the order of oxalic acid salt > cocrystals > drug	[65]
Lamotrigine	Cinnamic acid Ferulic acid Salicylic acid Vanillic acid	Poor flow properties and capping	Flow (angle of repose) and compression properties	All cocrystals except with ferulic acid showed improved flow properties; capping exhibited by cocrystal with salicylic acid was weak than that of ferulic acid	[66]
Paracetamol	5-nitroisophthalic acid	Poor mechanical properties	Tabletability – tensile strength	Superior tabletability of cocrystal than the drug	[67]
Sulfadimidine	4-aminosalicylic acid	To investigate the cocrystal habit engineering effect on compaction properties	Density and Carr’s compressibility index	Crystal habit engineering of cocrystals leads to improved flow properties	[68]
Adefovir Dipivoxil	Stearic acid	To investigate the enhanced powder properties of cocrystal	Compressibility (powder rheology analysis), Heckel analysis	Tabletability enhanced due to altered crystal habit by coformer	[69]

**Table 3 pharmaceutics-15-01161-t003:** Examples of studies reporting on coformers’ effect on stabilization of drug molecules in cocrystals.

Drug	Coformer(s)	Stability Issue of Drug	Parameter Assessed	Inference	Reference
Caffeine	Maleic acid Oxalic acid Glutaric acid Malonic acid	Crystalline powder of anhydrous caffeine transforms to caffeine hydrate at high RH	Physical stability at storage conditions of 0, 43, 75, and 98% RH up to 7 weeks.	No cocrystal hydrates have been found. Only oxalic acid-cocrystal exhibited physical stability till 7 weeks. The rest of cocrystals dissociated during storage.	[71]
Nitrofurantoin	4-hydroxybenzoic acid	Photosensitive and physicochemically unstable	Physical, chemical, and photostability at different conditions for 13 weeks	Improved physicochemical and photostability compared to pure drug	[73]
Adefovir dipivoxil	Saccharin Nicotinamide	Degradation by hydrolysis and dimerization during storage	Chemical stability at 60 °C (60% RH) and 40 °C (75% RH) for a month	Saccharin cocrystal was stable for one month whereas nicotinamide was not stable	[22]
Temozolomide	Salicylic acid Oxalic acid d,l-maleic acid Succinic acid d,l-tartaric acid	Spontaneous degradation during storage under normal conditions and is stable at pH < 5 but labile at pH > 7	Chemical stability at 40 °C and 75% RH for 28 weeks	Inhibited the hydrolytic degradation of the drug as cocrystal by providing the acidic environment with organic acid coformers	[74]
Tranilast	Urea Nicotinamide	Photochemically unstable	Subjected to 25 °C and 60% for 96 h	Photostability improved after the formation of cocrystal due to the increase in the distance between the drug molecules in the cocrystal	[75]
Acyclovir	Fumaric acid Malonic acid Tartaric acid	Hydration of drug during storage	Physical stability at storage conditions of 0%, 43%, 75%, and 98% RH for 3 weeks	Cocrystal showed improved stability except with tartaric acid	[53]
Etoricoxib	Suberic acid Glutaric acid Adipic acid Caprolactam	Hemihydrate conversion of drug during manufacturing or upon exposure to moisture (30 min)	Exposed to water (slurry) conditions for hydration	Formed stable cocrystals by replacing the water molecule in the crystal lattice	[72]
Epalrestat	Betaine	Photo instability	Subjected to 25 °C for 24 h	Improved photostability of cocrystal due to decreased reaction cavity	[21]
Isoniazid	Vanillic acid Caffeic acid	Reaction of isoniazid and rifampicin in fixed-dose combination	Physical stability of isoniazid in accelerated conditions	Stronger hydrogen bond interaction and cyclic O-H···O synthon in the crystal structure stabilized the cocrystal.	[76]
Flucytosine	Gallic acid Glutaric acid	Susceptible to hydration	Subjected to 70–75% RH and 90–95% RH at ambient temperature	Stable cocrystal may be due to strong acid–amide heterosynthon between drug and coformer	[77]

**Table 5 pharmaceutics-15-01161-t005:** Reported coformers in literature used in the formation of cocrystals.

API Name	Coformer/API Name	Cocrystals	API:Coformer Ratio	Reference
Temozolomide (TMZ)	Nicotinamide (NCT)	TMZ-NCT Cocrystal	2:1	[85]
	Isonicotinamide (INA)	TMZ-INA Cocrystal	2:1	
	Pyrazinamide (PYZ)	TMZ-PYZ Cocrystal	1:1	
	Saccharin (SAC)	TMZ-SAC Cocrystal	2:1	
	Caffeine (CAF)	TMZ-CAF Cocrystal	1:1	
Aripiprazole (ARI)	Orcinol (ORC)	ARI-ORC Cocrystal	1:1	[86]
	Catechol (CAT)	ARI-CAT Cocrystal	1:1	
	Resorcinol (RES)	ARI-RES Cocrystal	1:1	
	Phloroglucinol (PHL)	ARI-PHL Cocrystal	1:1	
Favipiravir (FAV)	4-hydroxybenzoic acid (4HBA)	FAV-4HBA Cocrystal	1:1	[87]
	p-aminobenzoic acid (PABA)	FAV-PABA Cocrystal	1:1	
	Ferulic acid (FRA)	FAV-FRA Cocrystal	1:1	
	Gallic acid (GA)	FAV-GA Cocrystal	1:1	
p-aminosalicylic acid (PAS)	Pyrazine (PYZ)	PAS-PYZ Cocrystal	1:1	[42]
	Pyrimidine (PYM)	PAS-PYM Cocrystal	1:1	
	Pyridazine (PDZ)	PAS-PDZ Cocrystal	2:1	
	Phenazine (PHZ)	PAS-PHZ Cocrystal	1:2	
	4,4′-dipyridyl disulfide (DPDS)	PAS-DPDS Cocrystal	1:1	
	4-cyanopyridine (4-CYP)	PAS-4-CYP Cocrystal (9)	1:1	
Curcumin (CUR)	Salicylic acid (SAA)	CUR-SAA Cocrystal	1:2	[88]
	Hydroxyquinol (HXQ)	CUR-HXQ Cocrystal	1:1, 1:2	
	Resorcinol (RNL)	CUR-RNL Cocrystal	1:1	
	Pyrogallol (PYG)	CUR-PYG Cocrystal	1:1	
	4,4′-bipyridine N, N′-dioxide (4,4 BPDO)	CUR-4,4 BPDO Cocrystal		
Salicylic acid (SA)	Benzamide (BZ)	SA-BZ Cocrystal	1:1, 1:2	
	Isonicotinamide (INA)	SA-INA Cocrystal	1:1, 2:1	
Carbamazepine (CMP)	4-aminobenzoic acid (4, ABA)	CMP-4 ABA Cocrystal	1:1, 2:1, 4:1	
Nicotinamide (NCT)	r-mandelic acid (r-MDLA)	NCT-r-MDLA Cocrystal	1:2, 1:1, 4:1	
Urea (UA)	Succinic acid (SA)	UA-SA Cocrystal	1:1, 2:1	
Indomethacin (IMC)	Saccharin (SAC)	IMC-SAC Cocrystal		
CL-20	Pyrazine (PYZ)	CL-20-PYZ Cocrystal		[23]
Theobromine (TBR)	Trimesic acid (TMSA)	TBR-TMSA Cocrystal	1:1	[89]
Theophylline (TPH)	Trimesic acid (TMSA)	TPH-TMSA Cocrystal	1:1	
Caffeine (CAF)	Trimesic acid (TMSA)	CAF-TMSA Cocrystal	1:2	
Theobromine (TBR)	Hemimellitic acid (HMLA)	TBR-HMLA Cocrystal	1:1	
Theophylline (TPH)	Hemimellitic acid (HMLA)	TPH-HMLA Cocrystal	1:1	
Caffeine (CAF)	Hemimellitic acid (HMLA)	CAF-HMLA Cocrystal	10:1	
5-Fluorouracil (5-FU)	Succinic acid (SA)	5-FU-SA Cocrystal	1:1	[90]
	Phenazine (PHZ)	5-FU-PHZ Cocrystal	2:1	
	Acridine (ACD)	5-FU-ACD Cocrystal	2:1	
	Benzoic acid (BA)	5-FU-BA Cocrystal	1:1	
	Malic acid (MA)	5-FU-MA Cocrystal	1:1	
	Cinnamic acid (CA)	5-FU-CA Cocrystal	1:1	
	4,4-bispyridylethene (4,4 BPYE)	5-FU-4,4 BPYE Cocrystal	4:1	
	p-aminopyridine (p-APY)	5-FU-p-APY Cocrystal		
Rivaroxaban (RVB)	Malonic Acid (MA) & Oxalic Acid (OA)	RVB-MA Cocrystals and RVB-OA Cocrystals	1:1, 2:1	[91]
	p-hydroxybenzoic acid (pHBA)	RVB-pHBA Cocrystals	1:1	[92]
	Isonicotinamide (INTA)	RVB-INTA Cocrystal	1:1	
	Nicotinamide (NTA)	RVB-NTA Cocrystal	1:1	
	2,4 dihydroxybenzoic acid (2,4 DHBA)	RVB-2,4 DHBA Cocrystals	1:1	
	Succinic acid (SA)	RVB-SA Cocrystals	1:1	
	Malonic Acid (MA)	RVB-MA Cocrystals	2:1	[93]
	Malonic Acid (MA)	RVB-MA Cocrystals	2:1	[94]
Lamotrigine (LTG)	Phthalimide (PTA)	LTG-PTA Cocrystals	1:1	[95]
	Succinic acid (SA)	LTG-SA Cocrystals		
	Pyromellitic diimide (PDA)	LTG-PDA Cocrystals	1:1	[96]
	Caffeine (CAF)	LTG-CAF Cocrystals	2:1	
	Isophthaldehyde (IPA)	LTG-IPA Cocrystals	1:1	
	Glutarimide (GTA)	LTG-GTA Cocrystals	1:1	[97]
	Phenobarbital (PBT) (Multi drug cocrystals)	LTG-PBT Cocrystals	1:3, 3:1	[98]
	2,2′-bipyridine (2,2 BPYD)	LTG-2,2 BPYD Cocrystals	1:1.5	[99]
	4,4′-bipyridine (4,4 BPYD)	LTG-4,4 BPYD Cocrystals	2:1	
	Nicotinamide monohydrate (NTAM)	LTG-NTAM Cocrystal	1:1:1	[100]
	Acetamide (ACT)	LTG-ACT Cocrystal	1:1	
	Acetic acid (ATA)	LTG-ATA Cocrystal	1:3	
	4-hydroxy-benzoic acid (4 HBA)	LTG-4 HBA Cocrystal	1:1	
	Saccharin (SAC)	LTG-SAC Cocrystal	1:1	
Etodolac (ETD)	4-amino benzoic acid (4 ABA)	ETD-4 ABA Cocrystal	1:1	[101]
	Glutaric acid (CA)	ETD-GA Cocrystal	1:2	[102]

**Table 6 pharmaceutics-15-01161-t006:** List of commercialized cocrystals.

Drug Name	Approval	Components	Dosage Form	Indication	Manufacturer	Ref. No.
Depakote^®^	U.S. FDA 1983	Valproic acid + Valproate sodium	Tablet, Capsule	Epilepsy	Abbott Laboratories, Illinois, United States	[15,16,103,104,105,106]
Entresto^®^	U.S. FDA 2015	Sacubitril sodium + Valsartan sodium	Tablet	Heart failure	Novartis, Basel, Switzerland	[15,16,107]
Suglat^®^	Japan 2014	Ipragliflozin + L-proline	Tablet	Diabetes	Kotobuki Pharmaceuticals, Nishina, Shizuoka, Japan and Astellas Pharma, Tokyo, Japan	[15,16,105,108]
Steglatro^®^	U.S. FDA 2017	Ertugliflozin + L-pyroglutamic acid	Tablet	Diabetes	Pfizer, New York, United States	[15,105,109,110]
Lexapro^®^	U.S. FDA 2002	Escitalopram oxalate + Oxalic acid	Tablet	Anxiety and depression	Allergan Inc., Dublin, Ireland	[15,16,111,112]
ESIX-10^®^	U.S. FDA 2009	Escitalopram oxalate + Oxalic acid	Tablet	Anxiety and depression	Sag Health Science Pvt Ltd., New Delhi, India	
Beta-chlor^®^	U.S. FDA 1963	Chloral hydrate + Betaine	Tablet	Sedation	Mead Johnson, Illinois, United States	[15,105]
Cafcit^®^	U.S. FDA 1999	Caffeine + Citric acid	Injection	Infantile apnoea	Hikma Pharmaceuticals Plc, London, United Kingdom	[16,105,113]
Zafatek^®^	Japan 2015	Trelagliptin + Succinic acid	Tablet	Diabetes	Takeda Pharmaceutical Company Limited, Tokyo, Japan	[16,114]
Lamivudine/zidovudine Teva ^®^	EMA 2011	Lamivudine + Zidovudine	Tablet	HIV infection	Teva Pharma B.V., Tel Aviv-Yafo, Israel	[16,115,116,117]
Abilify ^®^	U.S. FDA 2002	Aripiprazole + Fumaric acid	Tablet	Schizophrenia	Otsuka Pharmaceuticals, Tokyo, Japan	[128,129,135]
Odomzo^®^	U.S. FDA 2015	Sonidegib + Phosphoric acid	Capsule	Basal Cell Carcinoma	Sun Pharma Global, Mumbai, India.	[118,119,120]
Mayzent^®^	U.S. FDA 2019	Siponimod + Fumaric acid	Tablet	Multiple Sclerosis	Novartis, Basel, Switzerland	[121,122,123]
Seglentis^®^	U.S. FDA 2021	Celecoxib + Tramadol	Tablet	Acute Pain	Kowa Pharmaceuticals, Alabama, United States	[122,124]
Dimenhydrinate	U.S. FDA 1982 (ANDA)	Diphenhydramine and 8-chlorotheophylline	Tablet	Motion sickness	Watson Laboratories Inc., New Jersey, United States	[17,125,126]
Ibrutinib fumaric acid cocrystals	Tentative approval	Ibrutinib + Fumaric acid	NA	Cancer	Teva Pharmaceutical Industries Ltd., Tel Aviv-Yafo, Israel	[15,127]
E-58425 (Clinical Trial Phase 3)	Approval Pending	Celecoxib and racemic tramadol hydrochloride	NA	Management of acute pain	Patented by Laboratorios Del., La Paz, Bolivia Development done by Enantia and Esteve, R&D, Spain	[128,129,130,131,132]
TAK-020 (Clinical Trial Phase 1)	Approval Pending	TAK-020 and Gentisic acid	NA	Rheumatoid arthritis	Takeda Pharmaceuticals, Tokyo, Japan	[128,129,133,134]

**Table 7 pharmaceutics-15-01161-t007:** Physical and chemical properties of carboxylic acid-based coformers.

Coformer	Fumaric Acid [136,137,138,139]	Oxalic Acid [91,140,141]	Succinic acid [142]	Citric Acid [143,144]
Physical state	Colorless crystalline solid	Colorless crystalline solid	Colorless, odorless white crystals	Colorless crystalline solid
Melting point	287 °C	189.5 °C	185–187 °C	153 °C
Solubility in solvents	Soluble in ethanol, concentrated sulfuric acid. Slightly soluble in ethyl ether, acetone. Insoluble in chloroform and benzene.	Very soluble in ethanol. Slightly soluble in ether. Insoluble in benzene, chloroform, and petroleum ether.	Slightly soluble in ethanol, ether, acetone, glycerin. Not soluble in benzene, carbon sulfide, carbon tetrachloride.	Freely soluble in ethanol. Insoluble in benzene, chloroform, carbon tetrachloride, toluene, and carbon disulfide.
Solubility in water	7 g/L (25 °C)	220 mg/mL (25 °C)	Soluble (71 mg/mL)	592 mg/mL (20 °C)
Molar mass	116.07	90.03	118.09	192.1
Density	1.64 g/cm^3^	1.9 g/cm^3^	1.56 g/cm^3^	1.66 g/cm^3^
pKa	3.03	1.2	4.24	2.79
No. of hydrogen bond donors	2	2	2	4
No. of hydrogen bond acceptors	4	4	4	7
Stability	Stable under ambient conditions	Stable under ambient conditions	Stable under ambient conditions	Moisture-sensitive

**Table 8 pharmaceutics-15-01161-t008:** Impact of fumaric acid as coformer on drugs.

Cocrystal	Method of Preparation	Impact on Solubility	Impact on Dissolution Rate	Impact on Bioavailability	Impact on Stability	References
Berberine–Fumaric acid (2:1)	Slurry method	~9.5-fold at 15 min ^#^	~3.75-fold	────	Reduced hygroscopicity	[7]
Promethazine hydrochloride–Fumaric acid (2:1)	Mechanochemistry	Improved	────	────	Improved	[146]
	Slow solvent evaporation					
Gabapentin-lactam–Fumaric acid (1:1)	Reaction crystallization method	Improved	────	────	────	[147,158]
Fluoxetine HCl–Fumaric acid (2:1)	Cooling crystallization	Increased ~2-fold	No improvement	────	────	[151]
Enoxacin–Fumaric acid (1:2) ^@^	Slow solvent evaporation	9.8-fold	8.9-fold	────	────	[150]
Meloxicam–Fumaric acid (1:1)	Liquid-assisted grinding	33–84% improvement	────	────	────	[159]
Sildenafil–Fumaric acid (1:2 and 1:3)	Slow solvent evaporation	Increased 5-fold	────	────	────	[156]
PEC–Fumaric acid (1:1)	Anti-solvent addition	Increased 4-fold	Improved	Improved	────	[160]
Glipizide–Fumaric acid (1:1)	Liquid-assisted grinding	Increased ~2.3-fold	Increased ~2-fold	Improved	────	[161]
Acyclovir–Fumaric acid (1:1)	Slow solvent evaporation & liquid-assisted grinding	Increased ~5.5-fold	Increased ~2-fold	────	Improved	[53]
Ketoconazole–Fumaric acid (1:1, 1:2, and 1:3)	Slow solvent evaporation	Increased ~1.6-fold	Improved ~1.65-fold	────	Improved	[157]
Acyclovir–Fumaric acid (1:1)	Dry grinding or co-grinding	Less effect	Increased ~2.2-fold	────	────	[162]
Ethenzamide–Fumaric acid (2:1)	Slow solvent evaporation	Increased 3.84-fold	Increased 1.71-fold	────	────	[163]
Fluconazole–Fumaric acid (1:1)	Slow solvent evaporation	Increased ~2.5-fold ^#^	Improved	────	Improved	[164]
Efavirenz–Fumaric acid (1:1)	Neat grinding	Increased ~26-fold	Increased ~2-fold	────	────	[165]

#—Equilibrium solubility; @—2.3-fold improvement in permeability was also reported.

**Table 9 pharmaceutics-15-01161-t009:** Impact of oxalic acid as a coformer on drugs.

Cocrystal	Method of Preparation	Impact on Solubility	Impact on Dissolution rate	Impact on Bioavailability	Impact on Stability	References
Promethazine HCl–Oxalic acid (2:1)	Slow solvent evaporation	Improved	────	────	Improved	[146]
Telmisartan–Oxalic acid	────	Increased 7-fold	Increased ~2.4-fold	────	────	[122,169]
Rebamipide–Oxalic acid (1:1)	Liquid-assisted grinding	Increased 7.29-fold	Increased 7.19-fold	Increased 1.6-fold	────	
Rivaroxaban–Oxalic acid (1:1)	Anti-solvent addition	Improved	Increased ~1.6-fold	Increased ~2.12-fold	────	[91]
Apixaban–Oxalic acid (4:3)	────	Increased approx. 2-fold	────	Enhanced 2.7-fold	────	[166]
Temozolomide–Oxalic acid (2:1)	────	────	────	────	Improved	[167]
Xanthotoxin–Oxalic acid (2:1)	Liquid assisted grinding & slow solvent evaporation	Increased 1.6-fold	Increased ~1.1-fold	────	Improved	[168]
Telmisartan–Oxalic acid (1:1)	Solvent-drop grinding & solvent evaporation method	Increased 11.7-fold	Increased ~7.2-fold	────	────	[170]
Caffeine–Oxalic acid (2:1)	Solvent precipitation and ultrasound-assisted solution cocrystallization	────	────	────	Improved	[171]
Glibenclamide–Oxalic acid (1:2)	Thermal method	Increased ~2.7-fold	Increased ~1.7-fold	────	────	[172]

**Table 10 pharmaceutics-15-01161-t010:** Impact of succinic acid as a coformer on drugs.

API-Coformer	Method of Preparation	Impact on Solubility	Impact on Dissolution Rate	Impact on Bioavailability	Impact on Stability	References
Itraconazole–Succinic acid	Liquid anti-solvent	────	F1—Achieved 50% release in 2 h	────	Improved	[178]
	Gas anti-solvent		F2—Achieved 92% release in 2 h			
Piperine–Succinic acid	Wet-milling	Increased 12.70-fold	Achieved 53.281% release in 1 h	────	Improved	[179]
Carbamazepine–Succinic acid (2:1)	Slurry crystallization	Improved	F1—Achieved 82% release in 1 h F2—Achieved 95% release in 1 h F3—Achieved 95% release in 1 h	Improved	Improved	[2]
Isoniazid–Succinic acid (2:1)	Slow solvent evaporation	────	Improved	────	────	[180]
Imidazopyridazine- Succinic acid (1:1)	Neat grinding method	Improved	Improved	────	────	[181]
Brexpiprazole–Succinic acid (1:1)	Solvent-drop grinding method	Increased 1.59-fold	────	────	────	[182]
Aripiprazole–Succinic acid	Hot melt extrusion (HME)	────	Improved	────	────	[183]
Ketoconazole–Succinic acid (1:1)	Reaction crystallization method	────	Decreased with the low pH of coformer	────	────	[184]
Abiraterone acetate–Succinic acid (2:1)	Solvent evaporation	────	Increased 4.7-fold	────	Improved	[185]
Acyclovir–Succinic acid (1:1)	Grinding method		Dissolution efficiency– 54.23% (grinding time 15 min)	────	────	[186]
	Slurry crystallization		Dissolution efficiency – 74.36% (solvent concentration 12 mL/g)	────	────	
Fluoxetine HCl–Succinic acid (2:1)	Slow solvent evaporation	Increased ~1.5-fold	Increased 3-fold	────	────	[151]

F1—Formulation 1, F2—Formulation 2, F3—Formulation 3.

**Table 11 pharmaceutics-15-01161-t011:** Impact of citric acid as a coformer on drugs.

Cocrystals	Method of Preparation	Impact on Solubility	Impact on Dissolution rate	Impact on Bioavailability	Impact on Stability	References
Rebamipide–Citric acid (1:1)	Liquid-assisted grinding	Increased 12.58-fold	Increased ~13.2-fold	Increased 2.5-fold	────	[169]
Metformin hydrochloride–Citric acid (1:1)	Solution crystallization, neat grinding, and liquid-assisted grinding	Increased 1–4-fold	────	Improved	────	[143]
Berberine chloride- Citric acid (1:1)	Liquid-assisted grinding	Improved	────	────	Improved	[6]
Theophylline–Citric acid (1:1)	Neat co-grinding	────	────	────	Improved	[189]
Dapagliflozin propanediol monohydrate–Citric acid (1:1)	Solution crystallization method	Improved	Increased ~1-fold	────	Improved	[190]
Pyrazinamide–Citric acid (1:1)	Slow solvent evaporation	Increased ~1.1-fold	Increased ~1.4-fold	────	────	[56]
Ethyl p-methoxycinnamate–Citric acid (1:1, 1:2, 1:3)	Liquid-assisted grinding	Increased 1.4-fold	────	────	────	[192]
Simvastatin–Citric acid (1:1)	Liquid-assisted grinding, slow solvent evaporation	Increased 1.4–3-fold	────	Improved	Improved	[193]
Nefiracetam–Citric acid (2:1)	Slow solvent evaporation	Increased ~1.4-fold	Improved	────	────	[195]
Ritonavir–Citric acid (1:2)	Dry grinding method	Improved	Improved	────	────	[196]
Nitrofurantoin–Citric acid (1:1)	Liquid-assisted grinding	Improved	────	────	Improved	[197]
Praziquantel–Citric acid (1:1)	Liquid-assisted grinding	Increased ~2- 4-fold	Improved	────	────	[198]

**Table 12 pharmaceutics-15-01161-t012:** Role of coformers on the physicochemical properties of drug.

S. No.	Parameter	Fumaric Acid	Oxalic Acid	Succinic Acid	Citric Acid
1.	Solubility	Improved	Improved	Improved	Improved
2.	Dissolution rate	Improved	Improved	Improved	Improved
3.	Permeability	Improved	No Impact	No Impact	No Impact
4.	Bioavailability	Improved	Improved	Improved	Improved
5.	Stability	Improved	Improved	Improved	Improved
6.	No. of commercialized cocrystals	Two	Two	One	One

## Data Availability

The data presented in this study are available in the article and Appendix A.

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
