# Peer review of "Cocrystals by Design: A Rational Coformer Selection Approach for Tackling the API Problems"

_pharmaceutics, 2023, doi:10.3390/pharmaceutics15041161_

Round 1

Reviewer 1 Report

The review "Cocrystals by Design: A Rational Coformer Selection Approach for Tackling API Problems" by Maan Singh et al. summarizes the coformers used to improve the physicochemical properties of existing APIs. Overall, the text is readable and the manuscript is of very good quality, with a length of 49 pages, 192 references, tables, and figures. The review is quite interesting, and the amount of references, figures, and tables corresponds to the length of the text. The manuscript is suitable for publication, and the recommendations below are for improving the "completeness" of the subject. The authors may choose not to increase or supplement the review with the suggested additions. The recommendations are for some revisions, which could be minor or major.

The structure of the review is as shown below. However, it may not match the provided title. The main point that needs to be addressed is the lack of discussion on experimental screening techniques for conformer/cocrystal formation. Although Fig. 1 shows some techniques, from a practical point of view, it is not clear which screening and preparation techniques are suitable for cocrystal formation. Therefore, I suggest adding such a discussion after the "Comparison of Coformers" section, which is extremely short and needs expansion.

A section on drug-drug cocrystals/coformers is also lacking, although some examples of this type are given. In the "Most Popular Coformers Utilized in Cocrystal-Based Marketed Formulations" section, only four examples are shown.

The overall feel of the manuscript is an overview of publications related to cocrystal/coformers, rather than a review. The selection of coformers was based on the literature, specifically "the most popular coformers," as stated by the authors.

However, it is unclear on what scientific, pharmaceutical, statistical, or other criteria this selection was made. In p.5, the leading coformer is benzoic acid and it is omitted from the selection. This omission should be corrected or explained. (The review paper selected four coformers, fumaric acid, oxalic acid, succinic acid, and citric acid, based on reported scientific studies.)

The conclusions promoting the use of fumaric acid as a panacea coformer in API need to be ascertained.

1. Introduction

2. Selection of conformer

2.1 CSD

2.2 Hydrogen-Bond Rules

2.3 pKa rule

3. Coformers impact on pharmaceutical attributes

3.1. Role of coformers in solubility, dissolution, and bioavailability

3.2. Role of coformers in improving the mechanical properties of drug molecule

3.3. Role of coformers in stabilizing the drug molecule

3.4. Role of coformers in enhancing the permeability of cocrystals

4. Coformers reported in the literature

5. Coformers used in high demand

6. Commercially available drug products based on cocrystals

6.1- to 6.16

7. Most popular coformers utilized in cocrystal-based marketed formulations

7.1. Fumaric acid (FA)

7.2. Oxalic acid (OA)

7.3. Succinic acid

7.4. Citric acid

8. Comparison of coformers (those 4 from p/ 7.1-4) – to short, lack of explanation based on HB interactions, etc….

9. Patentability issue criteria and regulatory guidelines of pharmaceutical cocrystals

10. Conclusions

Minor

1. I would recommend to add the original reference for Etter rule e.g.:

Etter, M. C. Encoding and Decoding Hydrogen-Bond Patterns of Organic Compounds. Acc. Chem. Res. 1990, 23, 120−126.

2. Could you provide some references for Figure 2. (2) synthon.

3. Some recent references dealing with conformer/cocrystals have been omitted and need to be added in the text. I am providing just a short list

Wathoni, Nasrul, Wuri Ariestika Sari, Khaled M. Elamin, Ahmed Fouad Abdelwahab Mohammed, and Ine Suharyani. "A Review of Coformer Utilization in Multicomponent Crystal Formation." Molecules 27, no. 24 (2022): 8693.

Dhibar, Manami, Santanu Chakraborty, Souvik Basak, Paramita Pattanayak, Tanmay Chatterjee, Balaram Ghosh, Mohamed Raafat, and Mohammed AS Abourehab. "Critical Analysis and Optimization of Stoichiometric Ratio of Drug-Coformer on Cocrystal Design: Molecular Docking, In Vitro and In Vivo Assessment." Pharmaceuticals 16, no. 2 (2023): 284.

Bennett, Andrew J., and Adam J. Matzger. "Progress in Predicting Ionic Cocrystal Formation: the Case of Ammonium Nitrate." Chemistry–A European Journal (2023): e202300076.

Surov, Artem O., Anna G. Ramazanova, Alexander P. Voronin, Ksenia V. Drozd, Andrei V. Churakov, and German L. Perlovich. "Virtual Screening, Structural Analysis, and Formation Thermodynamics of Carbamazepine Cocrystals." Pharmaceutics 15, no. 3 (2023): 836.

Haneef, Jamshed, Mohd Amir, Nadeem Ahmed Sheikh, and Renu Chadha. "Mitigating Drug Stability Challenges Through Cocrystallization." AAPS PharmSciTech 24, no. 2 (2023): 62.

Point 3.4 Permeability ?

Some places the drugs are with first letter capitalized on others no ( Epalrestat / epalrestat ) . Please check.

On Table 1 you have only one decrease and it is 0.21 fold . Can you check for additional decrease examples?

Reviewer 2 Report

This manuscript is a comprehensive review article on cocrystals like fumaric and citric acids and their role in the development and formulation of certain difficult to administer drugs. In general, this reviewer had no technical or scientific issues with the manuscript and supports its publication after a minor revision suggestion: 

1. The conclusion section should be better written that should follow the conclusions from all the chapters of the review article. Please rewrite and make it better. 

2. It is a bit odd to have a supporting info in a review article. I would not make a SI for a review article. 

Reviewer 3 Report

The review article of Maan Singh et al. " Cocrystals by design: A rational coformer selection approach for tackling the API problems" contains an exhaustive review of conformer selection approach.

Introduction section appropriately states the background of the problem.

The review is well illustrated.

Tools for the selection of the appropriate coformers for cocrystals are highlited and described.

Commercialized products in addition to the scientific literature are used by the authors to underline the role of coformers in cocrystallization process.

As a major conclusion the authors marked fumaric acid to have benefits towards the enhancement of all the properties of interest (solubility, dissolution rate, permeability, and stability) of different APIs.

The review is appropriate to be published in Pharmaceutics.

At the same time, I have some comments, suggestions and corrections to be addressed by the authors:

Figure 1

Freeze-drying (liophilization) method of cocrystals preparation is missed. Add the information in the text and in Figure with the appropriate references, for ex. Drozd et al., 2022, doi:10.3390/pharmaceutics14051107, Karimi-Jafari et al., 2018, doi:10.1021/acs.cgd.8b00933.

Line 221

"the same" instead of "same"

Line 223

In the sentence " The solubility and dissolution improvement of coformer ……." Should be " The solubility and dissolution improvement of API ……."

Line 227-228

"There have been numerous examples in literature of such cases." Should be rewritten: " There have been numerous examples of such cases in the literature."

Figure 8

On the right "conformer" should be replaced with "coformer"

Tables 5 and S1

It is unclear why two tables with the same titles are given? Explain

Line 373-374

"….as depicted in the Figure 11." should be replaced with "….as depicted in Figure 11."

Line 508-509

"…. are enlist in the Table 6." should be replaced with "…. are enlist in Table 6."

Line 588-589

Since it is indicated in the text that "some reported cocrystals with fumaric acid as coformer showing improvement in terms of solubility, dissolution rate, permeability……" it is surprising that the permeability column is missed in Table 8. Explain please

Line 716-717

Check the sentence " Deng et al. [183] reported that dapagliflozin possess the stability problem at high temperature and also having the hygroscopicity issues." for style.

Round 2

Reviewer 1 Report

The  authors of the Review “Cocrystals by design: A rational coformer selection approach for tackling the API problems” responded  appropriately to most of the  concerns. The only point that is “ambiguous” remains the way the authors selected  "the most popular coformers". In the response the authors state “based on the number of approved formulations available in the market” but that opposes to the “very huge amount of information”. Nevertheless, the manuscript has been improved and its present form is suitable for  publication. Thus the recommendation is to  accept in its  present form.